# Entropy Accumulation Under Post-Quantum Cryptographic Assumptions

**DOI:** 10.3390/e27080772

**Published:** 2025-07-22

**Authors:** Ilya Merkulov, Rotem Arnon

**Affiliations:** Department of Physics of Complex Systems, Weizmann Institute of Science, Rehovot 7610001, Israel; rotem.arn@weizmann.ac.il

**Keywords:** device independent, post-quantum cryptography, entropy accumulation, randomness certification, quantum information theory

## Abstract

In device-independent (DI) quantum protocols, security statements are agnostic to the internal workings of the quantum devices—they rely solely on classical interactions with the devices and specific assumptions. Traditionally, such protocols are set in a non-local scenario, where two non-communicating devices exhibit Bell inequality violations. Recently, a new class of DI protocols has emerged that requires only a single device. In this setting, the assumption of no communication is replaced by a computational one: the device cannot solve certain post-quantum cryptographic problems. Protocols developed in this single-device computational setting—such as for randomness certification—have relied on ad hoc techniques, making their guarantees difficult to compare and generalize. In this work, we introduce a modular proof framework inspired by techniques from the non-local DI literature. Our approach combines tools from quantum information theory, including entropic uncertainty relations and the entropy accumulation theorem, to yield both conceptual clarity and quantitative security guarantees. This framework provides a foundation for systematically analyzing DI protocols in the single-device setting under computational assumptions. It enables the design and security proof of future protocols for DI randomness generation, expansion, amplification, and key distribution, grounded in post-quantum cryptographic hardness.

## 1. Introduction

The fields of quantum and post-quantum cryptography are rapidly evolving. In particular, the device-independent (DI) approach for quantum cryptography is being investigated in different setups and for various protocols. Consider a cryptographic protocol and a physical device that is being used to implement the protocol. The DI paradigm treats the device as untrusted and possibly adversarial. This means that in proving security, one must assume the device may have been prepared by an adversary. Only limited, well-defined assumptions regarding the inner workings of the device are placed. In such protocols, the honest party, called the verifier here, interacts with the untrusted device in a black-box manner, using classical communication. The security proofs are then based on properties of the transcript of the interaction, i.e., the classical data collected during the execution of the protocol, and the underlying assumptions.

The most well-studied DI setup is the so-called “non-local setting” [1,2]. There, the protocols are implemented using (at least) *two* untrusted devices, and the assumption made is that the devices cannot communicate between themselves during the execution of the protocol (or parts of it). In recent years, another variant has been introduced: instead of working with two devices, the protocol requires only a *single* device and the no-communication assumption is replaced by an assumption regarding the computational power of the device. More specifically, one assumes that during the execution of the protocol, the device is unable to solve certain *computational problems*, such as Learning With Errors (LWE) [3], which are believed to be hard for a quantum computer. (The exact setup and assumptions are explained in Section 3). Both models are DI, in the sense that the actions of the quantum devices are uncharacterized. Figure 1 schematically presents the two scenarios.

As in practice it might be challenging to assure that two quantum devices do not communicate, as required in the non-local setting, the incentive to study what can be achieved using only a single device is high. Indeed, after the novel proposals made in [4,5] for DI protocols for the verification of computation and randomness certification, many more protocols for various tasks and computational assumptions were investigated; see, e.g., [6,7,8,9,10,11,12,13,14,15,16]. Experimental works also followed, aiming at verifying quantum computation in the model of a single computationally restricted device [17,18]. With this research agenda advancing, more theoretical and experimental progress is needed—a necessity before one can estimate whether this avenue is of relevance for future quantum technologies or of sheer theoretical interest.

In this work we are interested in the task of generating randomness (Unless otherwise written, when we discuss the generation of randomness, we refer to a broad family of tasks: randomness certification, expansion, and amplification, as well as, potentially, quantum key distribution. In the context of the current work, the differences are minor). We consider a situation in which the device is prepared by a quantum adversary and then given to the verifier (the end user or costumer). The verifier wishes to use the device in order to produce a sequence of random bits. In particular, the bits should be random also from the perspective of the quantum adversary. The protocol defines how the verifier interacts with the device (also called the prover). It should be constructed to guarantee that the verifier aborts with high probability if the device is not trustworthy. Otherwise, it must certify that the bits produced by the device are random and unknown to the adversary.

Let us slightly formalize the above. The story begins with the adversary preparing a quantum state $\rho_{PE}^{\mathrm{in}}$; the marginal $\rho_{P}^{\mathrm{in}}$ is the initial state of the device given to the verifier while the adversary keeps the quantum register *E* for herself. In addition to the initial state $\rho_{P}^{\mathrm{in}}$, the device is described by the quantum operations, e.g., measurements, that it performs during the execution of the protocol. We assume that the device is a quantum polynomial time (QPT) device and thus can only perform efficient operations. Namely, the initial state $\rho_{P}^{\mathrm{in}}$ is of polynomial size and the operations can be described by a polynomial size quantum circuit; these are formally defined in Section 2. The verifier can perform only (efficient) classical operations. Together with the verifier, the initial state of all parties is denoted by $\rho_{PVE}^{\mathrm{in}}$. (In a simple scenario, one can consider $\rho_{PVE}^{\mathrm{in}}=\rho_{PE}^{\mathrm{in}}⊗\rho_{V}^{\mathrm{in}}$, i.e., the verifier is initially decoupled from the device and the adversary. This is, however, not necessarily the case in, for example, randomness amplification protocols [19]. We therefore allow for this flexibility with the above more general notation.) The verifier then executes the considered protocol using the device.

All protocols consist of what we call “test rounds” and “generation rounds”. The goal of a test round is to allow the verifier to check that the device is performing the operations that it is asked to apply by making it pass a test that only certain quantum devices can pass. The test and its correctness are based on the chosen computational assumption. (In the non-local setting, the test is based on the violation of a Bell inequality, or winning a non-local game with sufficiently high winning probability. Here the computational assumption “replaces” the Bell inequality.) For example, in [5], the cryptographic scheme being used is “Trapdoor Claw-Free Functions” (TCF)—a family of pairs of injective functions $f_{0},f_{1}:{\left\{0,1\right\}}^{n}\to {\left\{0,1\right\}}^{n}$. Informally speaking, it is assumed that, for every image *y*, (a) given a “trapdoor” one could classically and efficiently compute two pre-images $x_{0},x_{1}$ such that $f_{0}\left(x_{0}\right)=f_{1}\left(x_{1}\right)=y$; (b) without a trapdoor, even a quantum computer cannot come up with $x_{0},x_{1}$ such that $f_{0}\left(x_{0}\right)=f_{1}\left(x_{1}\right)=y$ (with high probability). While there does not exist an efficient quantum algorithm that can compute both pre-images $x_{0},x_{1}$ for a given image *y* without a trapdoor, a quantum device can nonetheless hold a *superposition* of the pre-images by computing the function over a uniform superposition of all inputs to receive $\sum_{y\in {\left\{0,1\right\}}^{n}}\left(|0,x_{0}\rangle +|1,x_{1}\rangle \right)|y\rangle$. These insights (and more—see Section 2.3 for details) allow one to define a test based on TCF such that a quantum device that creates $\sum_{y\in {\left\{0,1\right\}}^{n}}\left(|0,x_{0}\rangle +|1,x_{1}\rangle \right)|y\rangle$ can win while other devices that do not hold a trapdoor cannot. Moreover, the verifier, holding the trapdoor, will be able to check classically that the device indeed passes the test.

Let us move on to the generation rounds. In these rounds, the device produces the *output bits O*, which are supposed to be random. During the execution of the protocol, some additional information, such as a chosen public key for example (or whatever is determined by the protocol), may be publicly announced or leaked; we denote this *side information* by *S*.

After executing all the test and generation rounds, the verifier checks if the average winning probability in the test rounds is higher than some pre-determined threshold probability ω∈(0,1). If it is, then the protocol continues and otherwise aborts. Let the final state of the entire system, conditioned on not aborting the protocol, be $\rho_{|\Omega }$. To show that randomness has been produced, one needs to lower bound the conditional smooth min-entropy $H_{\min}^{\varepsilon }{\left(O|SE\right)}_{\rho_{|\Omega }}$ [20] (the formal definitions are given in Section 2). Indeed, this is the quantity that tightly describes the amount of *information-theoretically* uniform bits that can be extracted from the output *O*, given *S* and *E*, using a quantum-proof randomness extractor [21,22]. The focus of our work is to supply explicit lower bounds on $H_{\min}^{\varepsilon }{\left(O|SE\right)}_{\rho_{|\Omega }}$ in a general and modular way.

An important remark is in order before continuing. In our setting, the device is computationally bounded, but the output bits *O* must be secure against an unbounded adversary. This means that the adversary may retain her system *E* and later perform arbitrary operations on it, possibly using the side information *S*. Proving security in this setting implies that the computational assumption needs to hold only during the execution of the protocol. Once the output is generated, the security guarantee remains valid even if the computational assumption (such as LWE) is later broken. This property, known as “security lifting”, is a fundamental feature of all DI protocols based on computational assumptions.

### 1.1. Motivation of the Current Work

The setup of two non-communicating devices has naturally emerged from the study of quantum key distribution (QKD) protocols and non-local games (or Bell inequalities). As such, the quantum information theoretic toolkit for proving the security of protocols such as DIQKD, DI randomness certification, and alike was developed over many years and used for the analysis of numerous quantum protocols (see, for example, the survey [23]). The well-established toolkit includes powerful techniques that allow bounding the conditional smooth min-entropy $H_{\min}^{\varepsilon }{\left(O|SE\right)}_{\rho_{|\Omega }}$ mentioned above. Examples for such tools are the entropic uncertainty relations [24], the entropy accumulation theorem [25,26], and more. The usage of these tools allows one to derive quantitatively strong lower bounds on $H_{\min}^{\varepsilon }{\left(O|SE\right)}_{\rho_{|\Omega }}$, as well as handling realistically noisy quantum devices.

Unlike the protocols in the non-local setup, the newly developed protocols, for example, for randomness certification with a single device restricted by its computational power, are each analyzed using ad hoc proof techniques. On the qualitative side, such proofs make it harder to separate the wheat from the chaff, resulting in less modular and insightful claims. Quantitatively, the strength of the achieved statements is hard to judge—they are most likely not strong enough to lead to practical applications, and it is unknown whether this is due to a fundamental difficulty or a result of the proof technique. As a consequence, it is unclear whether such protocols are of relevance for future technology.

In this work, we show how to combine the information theoretic toolkit with assumptions regarding the computational power of the device. More specifically, we prove the lower bounds on the quantity $H_{\min}^{\varepsilon }{\left(O|SE\right)}_{\rho_{|\Omega }}$ by exploiting post-quantum cryptographic assumptions and quantum information-theoretic techniques, in particular the entropic uncertainty relation and the entropy accumulation theorem. Prior to our work, it was believed that such an approach cannot be taken in the computational setting (see the discussion in [5]). Once a bound on $H_{\min}^{\varepsilon }{\left(O|SE\right)}_{\rho_{|\Omega }}$ is proven, the security of the considered protocols then follows from our bounds using standard tools. The developed framework is general and modular. We use the original work of [5] as an explicit example; the same steps can be easily applied to, for example, the protocols studied in the recent works [14,15].

### 1.2. Main Ideas and Results

The main tool which is used to lower bound the conditional smooth min-entropy in DI protocols in the non-local setting is the entropy accumulation theorem (EAT) [25,27]. The EAT deals with sequential protocols, namely, protocols that proceed in rounds, one after the other, and in each round, some bits are being output. Roughly speaking, the theorem allows us to relate the total amount of entropy that accumulates throughout the execution of the protocol to, in some sense, an “average worst-case entropy of a single round” (see Section 2 for the formal statements). It was previously unclear how to use the EAT in the context of computational assumptions, and so [5] used an ad hoc proof to bound the total entropy.

The general structure of a protocol that we consider is shown in Figure 2. The initial state of the system is $\rho_{PVE}^{\mathrm{in}}$. The protocol, as mentioned, proceeds in rounds. Each round includes interaction between the verifier and the prover and, overall, can be described by an efficient quantum channel $M_{i}$ for round i∈[n]. The channels output some outcomes $O_{i}$ and side information $S_{i}$. In addition, the device (as well as the verifier) may keep the quantum and classical memory from previous rounds—this is denoted in the figure by the registers $R_{i}$. We remark that there is only one device and the figure merely describes the way that the protocol proceeds. That is, in each round i∈[n], the combination of the actions of the device and the verifier, together, defines the maps $M_{i}$. The adversary’s system *E* is untouched by the protocol. One could also consider more complex protocols in which the adversary’s information does change in some ways. This can be covered using the generalized EAT [26]. We do not do so in the current manuscript since all protocols so far fall in the above description, but the results can be extended to the setup of [26].

We are interested in bounding the entropy that accumulates by the end of the protocol: $H_{\min}^{\varepsilon }{\left(O|SE\right)}_{\rho_{|\Omega }}$, where $O=O_{1},\cdots ,O_{n}$ and similarly $S=S_{1},\cdots ,S_{n}$. The EAT tells us that, under certain conditions, this quantity is lower bounded, to the first order in *n*, by tn with *t* of the form(1)$$t=\min_{\sigma \in \Sigma (\omega )}H{\left(O_{i}|S_{i}E\right)}_{M_{i}\left(\sigma \right)}\;,$$

Here, $M_{i}$ denotes the quantum channel that describes the action of the *i*-th round of the protocol on the input state σ, including both the verifier’s interaction and the device’s response. The entropy is then computed with respect to the output distribution induced by this channel on the input state σ. The value ω is the average winning probability of the device in the test rounds, and Σ(ω) is the set of *all* (including inefficiently constructed) states of polynomial size that achieve the winning probability ω.

To lower bound the entropy appearing in Equation (Equation 1), one needs to use the computational assumption being made. This can be performed in various ways. In the current work, we show how statements about anti-commuting operators, such as those proven in [5,14,15], can be brought together with the entropic uncertainty relation [24], another tool frequently used in quantum information theory, to obtain the desired bound. In combination with our usage of the EAT, the bound on $H_{\min}^{\varepsilon }{\left(O|SE\right)}_{\rho_{|\Omega }}$ follows.

The described techniques are being made formal in the rest of the manuscript. We use [5] as an explicit example, also deriving quantitative bounds. We discuss the implications of the quantitative results in Section 4 and their importance for future works. Furthermore, our results can be used directly to prove the full security of the DIQKD protocol of [9].

The core technical result of this work is formalized in Theorem 2. This theorem provides a concrete, non-asymptotic lower bound on the conditional smooth min-entropy of the protocol output, conditioned on all side information including the adversary’s system. It gives a usable, round-scalable guarantee that applies to protocols based on noisy trapdoor claw-free functions, assuming only computational limitations on the device.

Informally, Theorem 2 states the following. If the device passes the test rounds with a sufficiently high success probability, then the output bits collected in the generation rounds contain min-entropy linear in the number of rounds, even conditioned on all classical and quantum side information.

We add that on top of the generality and modularity of our technique, its simplicity contributes to a better understanding of the usage of post-quantum cryptographic assumptions in randomness generation protocols. Thus, apart from being a tool for the analysis of protocols, we also shed light on what is required for a protocol to be strong and useful. For example, we can see from the explanation given in this section that the computational assumptions enter in three forms in Equation (Equation 1):The channels $M_{i}$ must be efficient.The states σ must be of polynomial size.Due to the minimization, the states σ that we need to consider may also be inefficient to construct, even though the device is efficient.

Points 1 and 2 are the basis when considering the computational assumption and constructing the test; in particular, they allow one to bound $H{\left(O_{i}|S_{i}E\right)}_{M_{i}\left(\sigma \right)}$, up to a negligible function η(λ), where λ is a security parameter defined by the computational assumption.

Point 3 is slightly different. The set over which the minimization is taken determines the strength of the computational assumption that one needs to consider. For example, it indicates that the protocol in [5] requires that the LWE assumption is hard even with a potentially inefficient polynomial-size advice state. This is a (potentially) stronger assumption than the “standard” formulation of LWE. Note that the stronger assumption is required even though the initial state of the device, $\rho_{P}^{\mathrm{in}}$, *is* efficient to prepare (i.e., it does not act as an advice state in this case). In [5], this delicate issue arises only when analyzing everything that can happen to the initial state throughout the entire protocol and conditioning upon not aborting. In the current work, we directly see and deal with the need for allowing advice states from the minimization in Equation (Equation 1).

We conclude our work with a discussion of what information is it that we allow our protocol to leak to the adversary via the side information register S. One obvious part of the side information S is the cryptographic key $K_{i}$, which was used to initiate the protocol, and the type of challenge (as was considered in [5]). Here we consider leakage of additional information to the adversary, and its consequent effect on the smooth min entropy $H_{\min}^{\varepsilon }{\left(O|SE\right)}_{\rho_{|\Omega }}$. With regard to the constituents of the side information, it is also important to note that *even* leakage of the trapdoor to the side information S is allowed. This should, naively, render the cryptographic primitives useless by undermining the computational assumptions, but in practice does not since we only demand that the computational assumptions hold, with respect to the device we are interacting with, *during* the execution of the protocol. Since the adversary *E* itself is unbounded and has access to the cryptographic key $K_{i}$, we can assume it computes the trapdoor.

### 1.3. Related Works

This work builds on [5] as a case study and draws on several of its results. We make use of the cryptographic primitive introduced in [5] (Section 3), as well as lemmas from [5] (Sections 6 and 7), which together define the core of the computational assumption underlying randomness generation.Our main advancement over this prior work is the ability to construct a von Neumann entropy bound for a single protocol round. This enables us to entirely replace the protocol-specific analysis in [5] (Section 8), a task that was conjectured to be beyond the reach of entropy accumulation techniques, by adapting the EAT framework of [28] to the computational setting.In Section 8 of [5], the authors develop a two-step analysis: they define an idealized device equipped with a fictitious projective measurement used to identify when the device lies in a “good subspace”, and then argue that the real protocol approximates the behavior of this idealized version. The entropy bound is derived manually, using a custom score function and round-by-round reductions that are specific to their protocol. While this approach is rigorous, it is mathematically involved and lacks generality.In contrast, our framework defines a per-round min-trade-off function and uses a general entropy accumulation argument to bound the total output entropy. This substitution yields a conceptually cleaner and modular proof, and allows us to derive concrete, non-asymptotic entropy bounds. This stands in contrast to the asymptotic nature of the results in [5], which do not quantify the amount of certified entropy for any finite choice of parameters.The absence of explicit bounds in [5] was later addressed by the same authors in [29], where a protocol variant generates *n* bits of randomness in a single round but without the ability to accumulate entropy across multiple rounds. As a result, achieving meaningful entropy guarantees in that setting requires choosing security parameters that are unreasonably large. This mirrors the trade-off in our work, where a large number of rounds is similarly required to achieve practically significant entropy rates.In [16], the authors investigate the certification of randomness on “NISQ” (Noisy Intermediate Scale Quantum) devices using the entropy accumulation theorem. The verification scheme proposed in that work is computationally hard. As a result, there is a tension between ensuring security and maintaining practicality: the security parameter must be large enough to make the underlying problem hard, yet small enough to keep verification feasible. Perhaps as a consequence, the computational assumptions described in that work are somewhat amorphous.

## 2. Preliminaries

Throughout this work, we use the symbol 1 as the identity operator and as the characteristic function interchangeably; the usage is clear due to context. We denote x←X when *x* is sampled uniformly from the set *X* or x←D when *x* is sampled according to a distribution D. The Bernoulli distribution with p(0)=γ is denoted as Bernoulli(1−γ) The Pauli operators are denoted by $\sigma_{x},\sigma_{y},\sigma_{z}$. The set {1,⋯,n} is denoted as [n].

### 2.1. Mathematical Background

**Definition** **1** (Negligible function)**.**
*A function $\eta :N\to R^{+}$ is to be negligible if for every c∈N there exists N∈N such that for every n>N, $\eta \left(n\right)<n^{-c}$.*


**Lemma** **1.**
*Let $\eta :N\to R^{+}$ be a negligible function. The function η(n)ln(1/η(n)) is also negligible.*


**Proof.** Assume, without loss of generality, the monotonicity and positivity of some negligible function η. Given c∈N, we wish to find N∈N such that ∀n>N, $\eta \left(n\right)\ln\left(1/\eta \left(n\right)\right)<n^{-c}$. We denote the following function:$$g\left(n\right):=\max\{\left.k\in N\right|\eta \left(n\right)<n^{-k}\}\;.$$This means$$\begin{array}{ccccccc} \; & n^{-(g(n)+1)} & \leq & \eta (n) & < & n^{-g(n)} & ,\\ \Rightarrow & -(g(n)+1)\ln n & \leq & \ln\eta (n) & < & -g(n)\ln n & ,\\ \Rightarrow & (g(n)+1)\ln n & \geq & \ln\left(1/\eta (n)\right) & > & g(n)\ln n & , \end{array}$$
yielding$$\eta \left(n\right)\ln\left(1/\eta \left(n\right)\right)<n^{-g(n)}\left(g\left(n\right)+1\right)\ln n=\frac{g(n)+1}{n^{g(n)-1}}\frac{\ln n}{n}<\frac{g(n)+1}{n^{g(n)-1}}=\frac{g(n)+1}{n^{g(n)-1-c}}\frac{1}{n^{c}}\;.$$By the negligibility property of η(n), the function g(n) diverges to infinity. Therefore, ∃N∈N such that ∀n>N$$\frac{g(n)+1}{n^{g(n)-1-c}}\leq 1\;,$$
and for all n>N
$$\eta \left(n\right)\ln\left(1/\eta \left(n\right)\right)<\frac{g(n)+1}{n^{g(n)-1-c}}\frac{1}{n^{c}}\leq \frac{1}{n^{c}}\;.$$
□

**Corollary** **1.**
*Let η be a negligible function and let h(x)=−xlogx−(1−x)log(1−x) be the binary entropy function. There exists a negligible function ξ for which the following holds:*

(2)
h(x−η(n))≥h(x)−ξ(n).



**Definition** **2** (Hellinger distance)**.**
*Given two probability distributions $P={\left\{p_{i}\right\}}_{i}$ and $Q={\left\{q_{i}\right\}}_{i}$, the Hellinger distance between P and Q is defined as*

$$H\left(P,Q\right)=\frac{1}{\sqrt{2}}\sqrt{\sum_{i}{\left(\sqrt{p_{i}}+\sqrt{q_{i}}\right)}^{2}}\;.$$



**Lemma** **2** (Jordan’s lemma extension)**.**
*Let $U_{1},U_{2}$ be two self-adjoint unitary operators acting on a Hilbert space H of a countable dimension. Let L be a normal operator acting on the same space such that $\left[L,U_{1}\right]=\left[L,U_{2}\right]=0$. There exists a decomposition of the Hilbert space into a direct sum of orthogonal subspaces $H={⊕}_{\alpha }H_{\alpha }$ such that for all α, $\dim\left(H_{\alpha }\right)\leq 2$ and given $|\psi \rangle \in H_{\alpha }$, all three operators satisfy $U_{1}|\psi \rangle ,U_{2}|\psi \rangle ,L|\psi \rangle \in H_{\alpha }$.*


Jordan’s lemma appears, among other places, in [2] (Appendix G.4). The sole change in the proof of the extension is in the choice of diagonalizing basis of the unitary operator $U_{2}U_{1}$. The chosen basis is now a mutual diagonalizing basis of $U_{2}U_{1}$ and *L*, which exists due to commutative relations.

**Lemma** **3.***Let Π,M,K be Hermitian projections acting on a Hilbert space H of a countable dimension such that [K,Π]=[K,M]=0. There exists a decomposition of the Hilbert space into a direct sum of orthogonal subspaces such that Π,M and K are a 2 by 2 block diagonal. In addition, in subspaces $H_{\alpha }$ of dimension 2,* Π *and M take the forms*
Πα=100Mα=cα2cαsαcαsαsα2*where $c_{\alpha }=\cos\theta_{\alpha },s_{\alpha }=\sin\theta_{\alpha }$ for some $\theta_{\alpha }$.*

**Proof.** Given a Hermitian projection Π, consider the unitary operator 2Π−1, satisfying ${\left(2\Pi -1\right)}^{2}=1$. Therefore, the operators 2Π−1,2M−1 satisfy the conditions for Lemma 2 and there exists the decomposition of H to a direct sum of orthogonal subspaces $H_{\alpha }$ such that in these subspaces,$$\begin{array}{cc} 2\Pi_{\alpha }-1_{\alpha }=\left(\begin{array}{cc} \; & 1\\ 1 \end{array}\right)\; & \; \end{array}2M_{\alpha }-1_{\alpha }=\left(\begin{array}{cc} \; & \omega \\ \bar{\omega } \end{array}\right)\;.$$One can recognize the operators as $\sigma_{x}$ and $\cos\theta \sigma_{x}+\sin\theta \sigma_{y}$, respectively, for some angle θ. Therefore, there exists a basis of $H_{\alpha }$ such that the operators are $\sigma_{z}$ and $\cos\phi \sigma_{z}+\sin\phi \sigma_{x}$, respectively, for some angle ϕ. Consequently, in this subspace, the projections take the form$$\begin{array}{c} \Pi_{\alpha }=\frac{1}{2}\left(\left(\begin{array}{cc} 1 & \\ \; & -1 \end{array}\right)+\left(\begin{array}{cc} 1 & \\ \; & 1 \end{array}\right)\right)=\left(\begin{array}{cc} 1 & \\ \; & 0 \end{array}\right)\\ M_{\alpha }=\frac{1}{2}\left(\left(\begin{array}{cc} \cos\phi & \sin\phi \\ \sin\phi & -\cos\phi \end{array}\right)+\left(\begin{array}{cc} 1 & \\ \; & 1 \end{array}\right)\right)=\left(\begin{array}{cc} c_{\alpha }^{2} & c_{\alpha }s_{\alpha }\\ c_{\alpha }s_{\alpha } & c_{\alpha }^{2} \end{array}\right) \end{array}\;,$$
with $\theta_{\alpha }=\phi /2$. □

**Lemma** **4** (Jensen’s inequality extension)**.**
*Let f:U→R be a convex function over a convex set $U\in R^{n}$ such that for every u∈U there exists a subgradient. Let ${\left(X_{i}\right)}_{i\in [n]}$ be a sequence of random variables with support over U. Then, $E\left[f\left(X_{1},\ldots ,X_{n}\right)\right]\geq f\left(E\left[X_{1}\right],\ldots ,E\left[X_{n}\right]\right)$.*


### 2.2. Tools in Quantum Information Theory

We state here the main quantum information theoretic definitions and techniques appearing in previous work, on which we build in the current manuscript.

**Definition** **3** (von Neumann entropy)**.**
*Let $\rho_{AB}$ be a density over the Hilbert space $H_{A}⊗H_{B}$. The von Neumann entropy of the $\rho_{A}$ is defined as*

$$H{\left(A\right)}_{\rho }=-\mathrm{Tr}\left(\rho_{A}\log\rho_{A}\right)\;.$$


*The conditional von Neumann entropy of A given B is defined to be*

$$H{\left(A|B\right)}_{\rho }=H{\left(AB\right)}_{\rho }-H{\left(B\right)}_{\rho }\;.$$



**Definition** **4**(Square overlap [24])**.** *Let* Π *and M be two observables, described by orthonormal bases ${\left\{|\pi_{i}\rangle \right\}}_{i}$ and ${\left\{|m_{j}\rangle \right\}}_{j}$ on a d-dimensional Hilbert space $H_{A}$. The measurement processes are then described by the completely positive maps*(3)$$P:\rho \to \sum_{i}\langle \pi_{i}|\rho |\pi_{i}\rangle |\pi_{i}\rangle \langle \pi_{i}|\;,\;M:\rho \to \sum_{j}\langle m_{j}|\rho |m_{j}\rangle |m_{j}\rangle \langle m_{j}|.$$*The square overlap of* Π *and M is then defined as*
$$c:=\max_{i,j}{\left|\langle \pi_{i}|m_{j}\rangle \right|}^{2}\;.$$

**Definition** **5.***Let M be an observable on a d-dimensional Hilbert space $H_{A}$ and let M be its corresponding map as appearing in Equation* (Equation 3). *Given a state $\rho \in H_{A}⊗H_{B}$, we define the conditional entropy of the measurement M given the side information B as*
(4)$$H{\left(M|B\right)}_{\rho }=H{\left(A|B\right)}_{\left(M⊗1_{B}\right)\left(\rho \right)}=H{\left(AB\right)}_{\left(M⊗1_{B}\right)\left(\rho \right)}-H{\left(B\right)}_{\left(M⊗1_{B}\right)\left(\rho \right)}\;.$$

**Lemma** **5**(Entropic uncertainty principle [24] (Supplementary—Corollary 2))**.** *Let* Π *and M be two observables on a Hilbert space $H_{P}$ and let c be their square overlap. For any density operator $\rho \in H_{V}⊗H_{P}⊗H_{E}$,*H(Π|E)≥log(1/c)−H(M|V).

In order to provide a clear understanding of the quantum uncertainty that arises from two measurements, as in Lemma 5, it is beneficial to examine the square overlap between those measurements (Definition 4). The Bloch sphere representation, which pertains to Hilbert spaces of two dimensions, offers a lucid illustration of this concept as shown in Figure 3.

**Lemma** **6.***Given two non-trivial Hermitian projections*(5)$$\Pi =\frac{1}{2}\left(1+\sigma_{z}\right)\;\;;\;\;M=\frac{1}{2}\left(1+\cos\left(\theta \right)\sigma_{z}+\sin\left(\theta \right)\sigma_{x}\right)\;,$$*acting on a two-dimensional Hilbert space H, the eigenvalues of the operator ΠMΠ+(1−Π)M(1−Π) are $\cos^{2}\left(\theta /2\right)$ and $\sin^{2}\left(\theta /2\right)$. In addition, the square overlap of the operators in Equation* (Equation 5) *is $c=\max\{\cos^{2}\left(\theta /2\right),\sin^{2}\left(\theta /2\right)\}$.*

**Lemma** **7**(Continuity of conditional entropies [30] (Lemma 2))**.** *For states ρ and σ on a Hilbert space $H_{A}⊗H_{B}$, if $\frac{1}{2}{∥\rho -\sigma ∥}_{1}\leq \epsilon \leq 1$, then*(6)$${|H\left(A|B\right)}_{\rho }-H{\left(A|B\right)}_{\sigma }|\leq \epsilon \log_{2}\left(|A|\right)+\left(1+\epsilon \right)h\left(\frac{\epsilon }{1+\epsilon }\right)\;.$$

**Lemma** **8**(Good-subspace projection [5] (Lemma 7.2))**.** *Let* Π *and M be two Hermitian projections on H and ϕ a state on H. Let ω=(Mϕ) and*$$\mu =|\frac{1}{2}-\mathrm{Tr}(M\Pi \phi \Pi )-\mathrm{Tr}\left(M\left(1-\Pi \right)\phi \left(1-\Pi \right)\right)|\;.$$*Let c∈(1/2,1]. Let $B_{j}$ be the orthogonal projection on the j’th 2×2 block as given in Lemma 3 and denote $c_{j}$ as the square overlap of* Π *and M in the corresponding block. Define* Γ *to be the orthogonal projection on all blocks such that the square overlap is bound by c:*
(7)$$\Gamma :=\sum_{j:c_{j}\leq c}B_{j}\;.$$
*Then,*

(8)
$$\mathrm{Tr}\left(\left(1-\Gamma \right)\phi \right)\leq \frac{2\mu +10\sqrt{1-\omega }}{{\left(2c-1\right)}^{2}}\;.$$



The above lemma was proven in [5]. Since it plays a crucial part in the work, we include the proof for completeness.

**Proof.** Using Jordan’s lemma we find a basis of H, in which(9)$$M={⊕}_{j}\left(\begin{array}{cc} a_{j}^{2} & a_{j}b_{j}\\ a_{j}b_{j} & b_{j}^{2} \end{array}\right)\;\;\;\;\mathrm{and}\;\;\;\;\Pi ={⊕}_{j}\left(\begin{array}{cc} 1 & \\ \; & 0 \end{array}\right)\;,$$
where $a_{j}=\cos\theta_{j}$, $b_{j}=\sin\theta_{j}$, for some angles $\theta_{j}$. Let Γ be the orthogonal projection on those two-dimensional blocks such that $\max\left(a_{j}^{2},b_{j}^{2}\right)\equiv c_{j}\leq c$. Note the following:
Γ commutes with both *M* and Π, but not necessarily with ϕ.Γ is the projection on 2×2 blocks where the eigenvalues of the operator ΠMΠ+(1−Π)M(1−Π) are in the range [1−c,c] as a consequence of Lemma 6.Assume that ω=1. This implies that ϕ is supported on the range of *M*. For any block *j*, let $B_{j}$ be the projection on the block and $p_{j}=\mathrm{Tr}\left(B_{j}\phi \right)$. It follows from the decomposition of *M* and Π to two-dimensional blocks and the definition of μ that (this is easily seen in the Bloch sphere representation):(10)$$|\sum_{j:c_{j}\leq c}p_{j}\left(a_{j}^{4}+b_{j}^{4}\right)+\sum_{j:c_{j}>c}p_{j}\left(a_{j}^{4}+b_{j}^{4}\right)-\frac{1}{2}|=\mu \;.$$Using that for *j* with $c_{j}>c$ such that $\max(a_{j}^{2},b_{j}^{2})\geq c$ and the fact that the polynomial 1−2x(1−x) is strictly rising in the regime x>1/2, we get$$a_{j}^{4}+b_{j}^{4}=1-2a_{j}^{2}b_{j}^{2}=1-2\max\left(a_{j}^{2},b_{j}^{2}\right)\left(1-\max\left(a_{j}^{2},b_{j}^{2}\right)\right)\geq 1-2c\left(1-c\right)=\frac{1}{2}+\frac{1}{2}{\left(2c-1\right)}^{2}\;.$$From Equation (Equation 10) and $a_{j}^{4}+b_{j}^{4}\geq 1/2$, it follows that$$\begin{array}{cc} 2\mu & =\sum_{j:c_{j}\leq c}p_{j}\left(1+{\left(2c_{j}-1\right)}^{2}\right)+\sum_{j:c_{j}>c}p_{j}\left(1+{\left(2c_{j}-1\right)}^{2}\right)-1\\ \; & =\sum_{j:c_{j}\leq c}p_{j}{\left(2c_{j}-1\right)}^{2}+\sum_{j:c_{j}>c}p_{j}{\left(2c_{j}-1\right)}^{2}\geq \sum_{j:c_{j}>c}p_{j}{\left(2c-1\right)}^{2}\;. \end{array}$$Consequently,(11)$$\mathrm{Tr}(\left(1-\Gamma \right)\phi )\leq \frac{2\mu }{{\left(2c-1\right)}^{2}}\;.$$This concludes the case where ω=1.Consider ω<1 and Tr(Mϕ)>0, as otherwise the lemma is trivial. Let $\phi^{\prime }=M\phi M/\mathrm{Tr}(M\phi )$. By the gentle measurement lemma [31] (Lemma 9.4.1),(12)$${∥\phi^{\prime }-\phi ∥}_{1}\leq 2\sqrt{1-\omega }\;.$$Using the definition of μ, it follows that$$|\frac{1}{2}-\mathrm{Tr}(M\Pi \phi^{\prime }\Pi )-\mathrm{Tr}(M\left(1-\Pi \right)\phi^{\prime }\left(1-\Pi \right))|\leq \mu +4\sqrt{1-\omega }\;.$$Following the same steps as those used for the case γ=0 yields an analogue of (Equation 11), with $\phi^{\prime }$ instead of ϕ on the left-hand side and $\mu +4\sqrt{1-\omega }$ instead of μ on the right-hand side. Applying again Equation (Equation 12), the same bound transfers to ϕ with an additional loss of $2\sqrt{1-\omega }$. □

A main tool that we are going to use is the entropy accumulation theorem (EAT) [25,32]. For our quantitative results, we used the version appearing in [25]; one can use any other version of the EAT in order to optimize the randomness rates, e.g., [32] as well as the generalization in [26]. We do not explain the EAT in detail; the interested reader is directed to [28] for a pedagogical explanation.

**Definition** **6** (EAT channels)**.**
*Quantum channels ${\left\{M_{i}:R_{i-1}\to R_{i}O_{i}S_{i}Q_{i}\right\}}_{i\in [n]}$ are said to be EAT channels if the following requirements hold:*

*1.* 
*${\left\{O_{i}\right\}}_{i\in [n]}$ are finite dimensional quantum systems of dimension $d_{O}$, and ${\left\{Q_{i}\right\}}_{i\in [n]}$ are finite-dimensional classical systems (RV). ${\left\{S_{i}\right\}}_{i\in [n]}$ are arbitrary quantum systems.*
*2.* 
*For any i∈[n] and any input state $\sigma_{R_{i-1}}$, the output state $\sigma_{R_{i}O_{i}S_{i}}=M_{i}\left(\sigma_{R_{i-1}}\right)$ has the property that the classical value $Q_{i}$ can be measured from the marginal $\sigma_{O_{i}S_{i}}$ without changing the state. That is, for the map $T_{i}:O_{i}S_{i}\to O_{i}S_{i}Q_{i}$ describing the process of deriving $Q_{i}$ from $O_{i}$ and $S_{i}$, it holds that ${\mathrm{Tr}}_{Q_{i}}\circ T_{i}\left(\sigma_{O_{i}S_{i}}\right)=\sigma_{O_{i}S_{i}}$.*
*3.* 
*For any initial state $\rho_{R_{0}E}^{\mathrm{in}}$, the final state $\rho_{\mathrm{OSX}E}=\left({\mathrm{Tr}}_{R_{n}}\circ M_{n}\circ \cdots \circ M_{1}\right)⊗1_{E}\rho_{R_{0}E}^{\mathrm{in}}$ fulfills the Markov chain conditions $O_{1},\cdots ,O_{i-1}↔S_{1},\cdots ,S_{i-1},E↔S_{i}$.*



**Definition** **7** (Min-trade-off function)**.***Let $\left\{M_{i}\right\}$ be a family of EAT channels and Q denote the common alphabet of $Q_{1},\ldots ,Q_{n}$. A differentiable and convex function $f_{\min}$ from the set of probability distributions p over Q to the real numbers is called a* min-trade-off function *for $\left\{M_{i}\right\}$ if it satisfies*$$f_{\min}\left(p\right)\leq \underset{\sigma_{R_{i-1}R^{\prime }}:M_{i}{\left(\sigma \right)}_{Q_{i}}=p}{\inf}{H(O_{i}|S_{i}R^{\prime })}_{M_{i}\left(\sigma \right)}$$*for all i∈[n], where the infimum is taken over all purifications of input states of $M_{i}$ for which the marginal on $Q_{i}$ of the output state is the probability distribution p.*

**Definition** **8.**
*Given an alphabet Q, for any $q\in Q^{n}$, we define the probability distribution ${\mathrm{freq}}_{q}$ over Q such that for any $\tilde{q}\in Q$,*

$${\mathrm{freq}}_{q}\left(\tilde{q}\right)=\frac{|\left\{i\in \left[n\right]:q_{i}=\tilde{q}\right\}|}{n}\;.$$



**Theorem 1** (Entropy Accumulation Theorem (EAT))**.***Let $M_{i}:R_{i-1}\to R_{i}O_{i}S_{i}Q_{i}$ for i∈[n] be EAT channels, let ρ be the final state,* Ω *an event defined over $Q^{n}$, $p_{\Omega }$ the probability of* Ω *in ρ and $\rho_{|\Omega }$ the final state conditioned on* Ω. *Let ε∈(0,1). For $\hat{\Omega }=\left\{{\mathrm{freq}}_{q}:q\in \Omega \right\}$ convex, $f_{\min}$ is a min-trade-off function for ${\left\{M_{i}\right\}}_{i\in [n]}$, and any t∈R such that $f_{\min}\left({\mathrm{freq}}_{q}\right)\geq t$ for any ${\mathrm{freq}}_{q}\in \hat{\Omega }$,*(13)$$H_{\min}^{\varepsilon }{\left(O|SE\right)}_{\rho_{|\Omega }}\geq nt-\mu \sqrt{n}\;,$$*where*
(14)$$\mu =2\left(\log\left(1+2d_{O}\right)+\left\lceil {∥\nabla f_{\max}∥}_{\infty }\right\rceil \right)\sqrt{1-2\log(\varepsilon \cdot p_{\Omega })}\;.$$

### 2.3. Post-Quantum Cryptography

**Definition** **9** (Trapdoor claw-free function family)**.**
*For every security parameter λ∈N, let $X\subseteq {\left\{0,1\right\}}^{w},Y$ and $K_{F}$ be finite sets of inputs, outputs, and keys, respectively. A family of injective functions*

$$F={\left\{f_{k,b}:X\to Y\right\}}_{k\in K_{F},b\in \left\{0,1\right\}}$$

*is said to be trapdoor claw-free (TCF) family if the following holds:*


*Efficient Function Generation: There exists a PPT algorithm ${\mathrm{Gen}}_{F}$ which takes the security parameter $1^{\lambda }$ and outputs a key $k\in K_{F}$ and a trapdoor t.*

*Trapdoor: For all keys $k\in K_{F}$, there exists an efficient deterministic algorithm ${\mathrm{Inv}}_{k}$ such that, given t, for all b∈{0,1} and x∈X, ${\mathrm{Inv}}_{k}\left(t,b,f_{k,b}\left(x\right)\right)=x$.*

*Claw-Free: For every QPT algorithm A receiving as input $\left(1^{\lambda },k\right)$ and outputting a pair $\left(x_{0},x_{1}\right)\in X^{2}$, the probability to find a claw is negligible, i.e., there exists a negligible function η for which the following holds:*

$$\underset{\overset{k\leftarrow K_{F}}{\left(x_{0},x_{1}\right)\leftarrow A\left(1^{\lambda },k\right)}}{\mathrm{Pr}}\left[f_{k,0}\left(x_{0}\right)=f_{k,1}\left(x_{1}\right)\right]\leq \eta \left(\lambda \right).$$




The noisy trapdoor claw-free (NTCF) family used in this work is a generalization of the standard TCF definition presented above. The NTCF primitive was introduced in [5] as part of the construction of cryptographic protocols secure against quantum adversaries. Although the cryptographic primitive underlying Protocol 1 is an NTCF, it is often more intuitive to reason about the simpler TCF structure from Definition 9, which captures the essential properties in the noise-free setting.

**Definition** **10**(Noisy NTCF family [5])**.** *For every security parameter λ∈N, let X,Y and $K_{F}$ be finite sets of inputs, outputs, and keys, respectively, and $D_{Y}$ the set of distributions over Y. A family of functions*$$F={\left\{f_{k,b}:X\to D_{Y}\right\}}_{k\in K_{F},b\in \left\{0,1\right\}}$$*is said to be a noisy trapdoor claw-free (NTCF) family if the following conditions hold:*
***Efficient Function Generation:*** 
*There exists a probabilistic polynomial time (PPT) algorithm ${\mathrm{Gen}}_{F}$ which takes the security parameter $1^{\lambda }$ and outputs a key $k\in K_{F}$ and a trapdoor t.****Trapdoor Injective Pair:*** 
*For all keys $k\in K_{F}$, the following two conditions are satisfied:**1.* 
*Trapdoor: For all b∈{0,1} and $x\neq x^{\prime }\in X,\;\mathrm{Supp}\left(f_{k,b}\left(x\right)\right)\cap \mathrm{Supp}\left(f_{k,b}\left(x^{\prime }\right)\right)=\emptyset$. In addition, there exists an efficient deterministic algorithm ${\mathrm{Inv}}_{k}$ such that for all b∈{0,1},x∈X and $y\in \mathrm{Supp}(f_{k,b}\left(x\right))$, ${\mathrm{Inv}}_{k}\left(t,b,y\right)=x$.*
*2.* 
*Injective Pair: There exists a perfect matching relation $R_{k}\subseteq X\times X$ such that $f_{k,0}\left(x_{0}\right)=f_{k,1}\left(x_{1}\right)$ if and only if $\left(x_{0},x_{1}\right)\in R_{k}$.*
***Efficient Range Superposition:*** 
*For every function in the family $f_{k,b}\in F$, there exists a function $f_{k,b}^{\prime }:X\to D_{Y}$ (not necessarily a member of F) such that the following hold:**1.* 
*For all $\left(x_{0},x_{1}\right)\in R_{k}$ and $y\in \mathrm{Supp}(f_{k,b}^{\prime }\left(x_{b}\right))$, ${\mathrm{Inv}}_{k}\left(t,b,y\right)=x_{b}$ and ${\mathrm{Inv}}_{k}\left(t,1-b,y\right)=x_{1-b}$.*
*2.* 
*There exists an efficient deterministic algorithm ${\mathrm{Chk}}_{k}$ (not requiring a trapdoor) such that*

$${\mathrm{Chk}}_{k}\left(b,x,y\right)=1_{y\in \mathrm{Supp}(f_{k,b}^{\prime }\left(x\right))}\;.$$

*3.* 
*There exists some negligible function η such that*

$$\underset{x\leftarrow X}{E}\left[H^{2}\left(f_{k,b}\left(x\right),f_{k,b}^{\prime }\left(x\right)\right)\right]\leq \eta \left(\lambda \right)$$


*where H is the Hellinger distance for distributions defined in Definition 2.*
*4.* 
*There exists a quantum polynomial time (QPT) algorithm ${\mathrm{Samp}}_{k,b}$ that prepares the quantum state*

$$|\psi^{\prime }\rangle =\frac{1}{\sqrt{\left|X\right|}}=\sum_{x\in X,y\in Y}\sqrt{(f_{k,b}^{\prime }\left(x\right))\left(y\right)}|x\rangle |y\rangle .$$

***Adaptive Hardcore Bit:*** 
*For all keys $k\in K_{F}$, the following holds. For some integer w that is a polynomially bounded function of λ, we have the following:**1.* 
*For all b∈{0,1} and x∈X, there exists a set $G_{k,b,x}\subseteq {\left\{0,1\right\}}^{w}$ such that ${\mathrm{Pr}}_{d\leftarrow {\left\{0,1\right\}}^{w}}\left[d\notin G_{k,b,x}\right]\leq \eta \left(\lambda \right)$ for some negligible function η. In addition, there exists a PPT algorithm that checks for membership in $G_{k,b,x}$ given k,b,x and the trapdoor t.*
*2.* 
*Let*

$$\begin{array}{c} \begin{array}{cc} H_{k}:= & \{\left(b,x_{b},d,d\cdot \left(x_{0}⊕x_{1}\right)\right)|b\in \left\{0,1\right\},\left(x_{0},x_{1}\right)\in R_{k},d\in G_{k,0,x_{0}}\cap G_{k,1,x_{1}}\}\;,\\ {\bar{H}}_{k}:= & \left\{\left(b,x_{b},d,e\right)|e\in \left\{0,1\right\},\left(b,x_{b},d,1-e\right)\in H_{k}\right\}\;. \end{array} \end{array}$$


*then for any QPT A and polynomial size (potentially inefficient to prepare) advice state ϕ independent of the key k, there exists a negligible function $\eta^{\prime }$ for which the following holds (we remark that the same computational assumption is being used in [5], even though the advice state is not included explicitly in the definition therein):*

(15)
$$|\underset{\left(k,t\right)\leftarrow Gen_{F(1^{\lambda })}}{\mathrm{Pr}}\left[A\left(k,\phi \right)\in H_{k}\right]-\underset{\left(k,t\right)\leftarrow Gen_{F(1^{\lambda })}}{\mathrm{Pr}}\left[A\left(k,\phi \right)\in {\bar{H}}_{k}\right]|\leq \eta^{\prime }\left(\lambda \right)\;.$$




**Lemma** **9**(Informal: existence of NTCF family [5] (Section 4))**.** *There exists an LWE-based construction of an NTCF family as in Definition 10. (As previously noted, [5] assumes that the LWE problem is hard also when given an advice state. Thus, though not explicitly stated, the construction in [5] (Section 4) is secure with respect to an advice state.)*


**Protocol 1** Entropy accumulation protocol.

**Arguments:**
  *D*—untrusted device with which the verifier can interact according to the described protocol.  F— NTCF  n∈N—number of rounds  γ∈(0,1]—expected fraction of test rounds  ω∈[0,1]—threshold winning rate  β∈(0,1)—pre-image test to equation test ratio
**Process:**
1:  Set all variables to the default value ⊥.2:  For every round i∈[n] do Steps 3–14.3:    Sample key and trapdoor $\left(K_{i},\tau_{i}\right)\leftarrow {\mathrm{Gen}}_{F}$ for the NTCF F.4:    Pass the key $K_{i}$ to the device *D* which in return outputs $Y_{i}\in Y$.5:    Verifier samples $G_{i}\leftarrow \mathrm{Bernoulli}\left(1-\gamma \right)$.6:    If $G_{i}=0$: Test round7:      Verifier samples $\;T_{i}\leftarrow \mathrm{Bernoulli}\left(\beta \right)$ and passes it to the device *D*.8:      If $T_{i}=0$, pre-image test. *D* outputs $(b_{i},x_{i})\in \left\{0,1\right\}\times X$.9:        Verifier sets10:          ${\hat{\Pi }}_{i}=b_{i}$ if ${\mathrm{Chk}}_{k_{i}}\left(b_{i},x_{i},Y_{i}\right)=1$ and 2 otherwise.11:          $W_{i}={\mathrm{Chk}}_{k_{i}}\left(b_{i},x_{i},Y_{i}\right)$.12:      If $T_{i}=1$, equation test. *D* outputs $\left(u_{i},d_{i}\right)\in \left\{0,1\right\}\times {\left\{0,1\right\}}^{w}$.13:        Verifier, using $\tau_{i}$, sets ${\hat{M}}_{i}=1$ if $\left(u_{i},d_{i}\right)$ is a valid response for the        challenge (0 otherwise) and $W_{i}={\hat{M}}_{i}$.14:    If $G_{i}=1$: Generation round15:      Verifier sets $T_{i}=0$, and passes $T_{i}$ to the device *D*.16:        *D* outputs $(b_{i},x_{i})\in \left\{0,1\right\}\times X$.17:        Verifier sets ${\hat{\Pi }}_{i}=b_{i}$ if ${\mathrm{Chk}}_{k_{i}}\left(b_{i},x_{i},Y_{i}\right)=1$ and 2 otherwise.18:  Abort if $\frac{1}{\gamma \cdot n}\sum_{j:G_{j}=0}W_{j}<\omega$.**Output:** ${\hat{\Pi }}_{j}$ for all *j* such that $G_{j}=1$.



Protocol 1: High-level description of the entropy accumulation protocol. In each round, the verifier interacts with an untrusted device by sending a challenge based on a noisy trapdoor claw-free function and receiving a response. The round is designated as either a test or generation round. In test rounds, the verifier checks whether the device behaves consistently with the structure of the function (either via a pre-image or equation check). In generation rounds, the verifier collects output bits intended to serve as randomness. The variable ${\hat{\Pi }}_{i}$ encodes the outcome of the round in a unified way: it stores a candidate output bit if the associated check passes, and a placeholder value (2) otherwise. This allows both test and generation rounds to be treated uniformly in the later entropy analysis.

## 3. Randomness Certification

Our main goal in this section is to provide a framework for lower bounding the entropy accumulated during the execution of a protocol involving a single quantum device. Protocol 1 specifies the randomness generation protocol used in our setting, which consists of *n* rounds of classical interaction between a verifier and a quantum device. The security of the protocol is based on the existence of a noisy trapdoor claw-free (NTCF) family $F=\{f_{b,k}:X\to Y\}$, constructed under the LWE assumption [3], following [5]. The protocol makes use of the keys sampled from F and a security parameter λ, with all definitions stated implicitly in terms of these.

In each round, the verifier samples a bit $G_{i}$ to determine whether the round is a test round ($G_{i}=0$) or a generation round ($G_{i}=1$). When $T_{i}=0$, the device produces a pair $\left(b_{i},x_{i}\right)$ in both types of rounds. The difference lies in how the verifier handles this response: In test rounds, the verifier uses the trapdoor to check whether the response is consistent with the challenge, contributing to a success/failure count. In generation rounds, no check is performed, and the output bit $b_{i}$ is retained as part of the protocol’s output. Correctness in generation rounds is inferred from the device’s performance in the test rounds.

We denote by ${\hat{\Pi }}_{i}$ a unified variable that stores either the output bit (when a check is passed or skipped) or a placeholder value otherwise. The final quantitative result of this section is a lower bound on the smooth min-entropy of the output bits $\hat{\Pi }$, conditioned on the adversary’s information and public transcript.

### 3.1. Single-Round Entropy

In this section, we analyze a single round of Protocol 1, presented as Protocol 2.


**Protocol 2** Single-round protocol.

**Arguments:**
  Classical verifier V  Quantum prover P represented by the device D=(ϕ,Π,M)  F—NTCF  β∈(0,1)
**Process:**
1:  V samples $\left(k,\tau \right)\leftarrow {\mathrm{Gen}}_{F}$ for the NTCF F.2:  V sends the key *k* to the prover.3:  P sends y∈Y to V.4:  V samples T←Bernoulli(β) and gives it to P.5:  If T=06:    P returns (b,x)∈{0,1}×X to V.7:    V sets $\hat{\pi }=b$ if ${\mathrm{Chk}}_{k}\left(b,x,y\right)=1$ and 2 otherwise. $W={\mathrm{Chk}}_{k}\left(b,x,y\right)$.8:  If T=19:    P returns $\left(u,d\right)\in \left\{0,1\right\}\times {\left\{0,1\right\}}^{w}$ to V.10:    V, using τ, sets W=1 if (u,d) is a valid response to the challenge.



Protocol 2: Description of the single-round protocol.

#### 3.1.1. One Round Protocols and Devices

Protocol 2, roughly, describes a single round of Protocol 1. The goal of Protocol 2 can be clarified by thinking about an interaction between a verifier V and a quantum prover P in an independent and identically distributed (IID) scenario, in which the device is repeating the same actions in each round. Upon multiple rounds of interaction between the two, V is convinced that the winning rate provided by P is as high as expected and therefore can continue with the protocol. We will, of course, use the protocol later on without assuming that the device behaves in an IID manner throughout the multiple rounds of interaction.

We begin by formally defining the most general device that can be used to execute the considered single round Protocol 2.

**Definition** **11**(General device [5])**.** *A general device is a tuple D=(ϕ,Π,M) that receives $k\in K_{F}$ as input and is specified by the following:*
*1.* 
*A normalized density matrix $\phi \in H_{D}⊗H_{Y}$.*

*$H_{D}$ is a polynomial (in λ) space, private to the device.*

*$H_{Y}$ is a space private to the device whose dimension is the same of the cardinality of Y.*

*For every y∈Y, $\phi_{y}$ is a sub-normalized state such that*

$$\phi_{y}=\left(1_{D}⊗\langle {y|}_{Y}\right)\phi \left(1_{D}⊗|{y\rangle }_{Y}\right).$$

*2.* 
*For every y∈Y, a projective measurement $M_{y}^{\left(u,d\right)}$ on $H_{D}$, with outcomes $\left(u,d\right)\in \left\{0,1\right\}\times {\left\{0,1\right\}}^{w}$ (The value w is determined by the NTCF construction. The string d is used in equation-test rounds as a witness that the verifier can check using the trapdoor. It does not play a direct role in the entropy analysis).*
*3.* 
*For every y∈Y, a projective measurement $\Pi_{y}^{\left(b,x\right)}$ on $H_{D}$, with outcomes (b,x)∈{0,1}×X.*



**Definition** **12** (Valid response)**.**
*Let y∈Y be a challenge derived from the NTCF family associated with key k.*

*1.* *A pair (b,x)∈{0,1}×X is called a* valid response to a pre-image test *if ${\mathrm{Chk}}_{k}\left(b,x,y\right)=1$. There are exactly two such valid responses for each challenge y, $\left(0,x_{0}\right)$ and $\left(1,x_{1}\right)$.**2.* *A pair $\left(u,d\right)\in \left\{0,1\right\}\times {\left\{0,1\right\}}^{w}$ is called a* valid response to an equation test *if $\left(b,x_{b},d,u\right)\in H_{k}$ for both b∈{0,1}.*
*Intuitively, this means that if $u=d\cdot (x_{0}⊕x_{1})$ holds, then the response is valid (up to some choices of d that the verifier always rejects, such as $d=0^{w}$).*


*We denote these sets of valid responses as $V_{y,0}$ and $V_{y,1}$, for the pre-image and equation tests, respectively. These conditions define the responses that the verifier accepts during test rounds of Protocol 1.*


**Definition** **13** (Efficient device)**.**
*We say that a device D=(ϕ,Π,M) is efficient if the following holds:*

*1.* 
*The state ϕ is a polynomial size (in λ) “advice state” that is independent of the chosen keys $k\in K_{F}$. The state might not be producible using a polynomial time quantum circuit (we say that the state can be inefficient).*
*2.* *The measurements* Π *and M can be implemented by polynomial size quantum circuit.*


The above definitions do not specify an honest strategy. Rather, they provide a formal abstraction that allows us to analyze the entropy generated by the device in a single round. This abstraction is crucial for our proof technique: it isolates a simplified description of the device in terms of a fixed state ϕ and two projective measurements Π and *M*, which correspond to the two possible test types in the protocol. By reducing to this formal model, we are able to apply tools such as the entropic uncertainty relation and construct a min-trade-off function compatible with the entropy accumulation theorem.

Importantly, this abstraction is not purely theoretical—it is designed to faithfully capture the observable behavior of real quantum devices, whether honest or adversarial, while making the analysis tractable. In particular, an honest device fits naturally within this formalism. For example, it may prepare a state of the form $\frac{1}{\sqrt{2}}\left(|0,x_{0}\rangle +|1,x_{1}\rangle \right)$, where $f_{0}\left(x_{0}\right)=f_{1}\left(x_{1}\right)=y$, and perform either a computational basis measurement (yielding (b,x)) or a Hadamard measurement (yielding (u,d)), depending on the challenge. Such a device is efficient according to Definition 13 and behaves in a way that satisfies the test criteria of the protocol with probability 1.

To ensure that the analysis remains sound even for dishonest or arbitrarily structured devices, we impose computational constraints on the device model. These constraints limit both its time and memory resources, and are essential for the validity of the cryptographic assumptions used in the protocol.

We emphasize that the device *D* is computationally bounded by both time steps and memory. This prevents pre-processing schemes in which the device manually goes over all keys and pre-images to store a table of answers to all the possible challenges, as such schemes demand exponential memory in λ.

The cryptographic assumption made on the device is the following. Intuitively, the lemma states that due to the hardcore bit property in Equation (Equation 15), the device cannot pass both pre-image and equation tests. Once it passes the pre-image test, trying to pass also the equation test results in two computationally indistinguishable states—one in which the device also passes the equation test and one in which it does not.

**Lemma** **10**(Computational indistinguishability [5] (Lemma 7.1))**.** *Let D=(ϕ,Π,M) be an efficient general device as in Definitions 11 and 14. Define a sub-normalized density matrix*$${\tilde{\phi }}_{YBXD}=\sum_{y\in Y}{\left|y\rangle \langle y\right|}_{Y}⊗\sum_{b\in \left\{0,1\right\}}{\left|b,x_{b}\rangle \langle b,x_{b}\right|}_{BX}⊗\Pi_{y}^{\left(b,x_{b}\right)}\phi_{y}\Pi_{y}^{\left(b,x_{b}\right)}\;.$$
*Let*

$$\begin{array}{c} \begin{array}{cc} \sigma_{0} & =\sum_{b\in \left\{0,1\right\}}{\left|b,x_{b}\rangle \langle b,x_{b}\right|}_{BX}⊗\sum_{\left(u,d\right)\in V_{y,1}}{\left|u,d\rangle \langle u,d\right|}_{U}⊗\left(1_{Y}⊗M_{y}^{\left(u,d\right)}\right){\tilde{\phi }}_{YD}^{\left(b\right)}\left(1_{Y}⊗M_{y}^{\left(u,d\right)}\right)\;,\\ \sigma_{1} & =\sum_{b\in \left\{0,1\right\}}{\left|b,x_{b}\rangle \langle b,x_{b}\right|}_{BX}⊗\sum_{\left(u,d\right)\notin V_{y,1}}1_{d\in {\hat{G}}_{y}}{\left|u,d\rangle \langle u,d\right|}_{U}⊗\left(1_{Y}⊗M_{y}^{\left(u,d\right)}\right){\tilde{\phi }}_{YD}^{\left(b\right)}\left(1_{Y}⊗M_{y}^{\left(u,d\right)}\right)\;, \end{array} \end{array}$$

*where $V_{y,1}$ is the set of valid responses to the equation test challenge y as defined in Definition 12. Then, $\sigma_{0}$ and $\sigma_{1}$ are computationally indistinguishable.*


The proof is given in [5] (Lemma 7.1). Note that even though the proof in [5] does not address the potentially inefficient advice state ϕ explicitly, it holds due to the non-explicit definition of their computational assumption.

We proceed with a reduction of Protocol 2 to a simplified one, Protocol 3. As the name suggests, it will be easier to work with the simplified protocol and devices when bounding the produced entropy. We remark that a similar reduction is used in [5]; the main difference is that we are using the reduction on the level of a single round, in contrast to the way it is used in [5] (Section 8) when dealing with the full multi-round protocol. Using the reduction in the single-round protocol instead of the full protocol helps in disentangling the various challenges that arise in the analysis of the entropy.


**Protocol 3** Simplified single-round protocol.

**Arguments:**
  Classical verifier V  Quantum prover P represented by the simplified device $\tilde{D}=\left(\phi ,\tilde{\Pi },\tilde{M}\right)$  β∈(0,1)
**Process:**
1:  V samples the random variable T←Bernoulli(β) which is then passed to P.2:  If T=03:    P measures $\tilde{\Pi }$ on the state ϕ for an outcome p∈{0,1,2}.4:  If T=15:    P measures $\tilde{M}$ on the state ϕ for an outcome m∈{0,1}.



Protocol 3: Description of the simplified single-round protocol.

**Definition** **14**(Simplified device [5] (Definition 6.4))**.** *A simplified device is a tuple $\tilde{D}=\left(\phi ,\tilde{\Pi },\tilde{M}\right)$ that receives $k\in K_{F}$ as input and specified by the following:*
*1.* 
*$\phi ={\left\{\phi_{y}\right\}}_{y\in Y}\subseteq \mathrm{Pos}\left(H_{D}\right)$ is a family of positive semidefinite operators on an arbitrary space $H_{D}$ such that $\sum_{y}\mathrm{Tr}\left({\tilde{\phi }}_{y}\right)\leq 1$;*
*2.* 
*$\tilde{\Pi }$ and $\tilde{M}$ are defined as the sets ${\left\{{\tilde{\Pi }}_{y}\right\}}_{y},{\left\{{\tilde{M}}_{y}\right\}}_{y}$, respectively, such that for each y∈Y, the operators, ${\tilde{M}}_{y}=\{{\tilde{M}}_{y}^{0},{\tilde{M}}_{y}^{1}=1-{\tilde{M}}_{y}^{0}\}$ and ${\tilde{\Pi }}_{y}=\left\{{\tilde{\Pi }}_{y}^{0},{\tilde{\Pi }}_{y}^{1},{\tilde{\Pi }}_{y}^{2}=1-{\tilde{\Pi }}_{y}^{0}-{\tilde{\Pi }}_{y}^{1}\right\}$, are projective measurements on $H_{D}$.*



**Definition** **15**(Simplified device construction [5])**.** *Given a general device as in Definition 11 D=(ϕ,Π,M), we construct a simplified device $\tilde{D}=\left(\phi ,\tilde{\Pi },\tilde{M}\right)$ in the following manner:*

*The device $\tilde{D}$ measures y∈Y like the general device D would.*

*The measurement $\tilde{\Pi }=\left\{{\tilde{\Pi }}_{y}^{0},{\tilde{\Pi }}_{y}^{1},{\tilde{\Pi }}_{y}^{2}\right\}$ is defined as follows:*
–
*Perform the measurement ${\left\{\Pi_{y}^{\left(b,x\right)}\right\}}_{b\in \left\{0,1\right\},x\in X}$ for an outcome (b,x).*
–
*If ${\mathrm{Chk}}_{k}\left(b,x,y\right)=1$, the constructed device returns b corresponding to the projection ${\tilde{\Pi }}_{y}^{b}\in \left\{{\tilde{\Pi }}_{y}^{0},{\tilde{\Pi }}_{y}^{1}\right\}$.*
–
*If ${\mathrm{Chk}}_{k}\left(b,x,y\right)=0$, the constructed device returns 2, corresponding to the projection ${\tilde{\Pi }}_{y}^{2}$.*

*The measurement $\tilde{M}=\left\{{\tilde{M}}_{y}^{0},{\tilde{M}}_{y}^{1}\right\}$ is defined as follows:*

$$\begin{array}{cc} {\tilde{M}}_{y}^{0}=\sum_{\left(u,d\right)\in V_{y,1}}M_{y}^{\left(u,d\right)}\;\;\;,\;\;\; & {\tilde{M}}_{y}^{1}=1-{\tilde{M}}_{y}^{0}\;, \end{array}$$


*where $V_{y,1}$ is valid answers for the equation test, meaning the outcome $\tilde{M}=0$ corresponds to a valid response (u,d) in the equation test.*



The above construction of a simplified device maintains important properties of the general device. Firstly, the simplified device fulfills the same cryptographic assumption as the general one. This is stated in the following corollary.

**Corollary** **2.**
*Given a general efficient device D=(ϕ,Π,M), a simplified device $\tilde{D}=\left(\phi ,\tilde{\Pi },\tilde{M}\right)$ constructed according to Definition 15 is also efficient. Hence, the cryptographic assumption described in Lemma 10 holds also for $\tilde{D}$ as well.*


Secondly, the entropy produced by the simplified device in the simplified single-round protocol, Protocol 3, is identical to that produced by the general device in single round protocol, Protocol 2. A general device executing Protocol 2 defines a probability distribution of $\hat{\pi }$ over {0,1,2}. Using the same general device to construct a simplified one, via Definition 15, leads to the same distribution for $\tilde{\Pi }$ when executing Protocol 3. This results in the following corollary.

**Corollary** **3.**
*Given a general efficient device D=(ϕ,Π,M) and a simplified device $\tilde{D}=\left(\phi ,\tilde{\Pi },\tilde{M}\right)$ constructed according to Definition 15, we have for all k,*

$$H{\left(\hat{\pi }|EY\right)}^{\mathrm{General}}=H{\left(\tilde{\Pi }|EY\right)}^{\mathrm{Simplified}}\;,$$

*where $H{\left(\hat{\pi }|EY\right)}^{\mathrm{General}}$ is the entropy produced by Protocol 2 using the general device D, and $H{\left(\tilde{\Pi }|EY\right)}^{\mathrm{Simplified}}$ is the entropy produced in Protocol 3 using the simplified device $\tilde{D}$. Both entropies are evaluated on the purification of the state ϕ.*


The above corollary tells us that we can reduce the analysis of the entropy created by the general device to that of the simplified one, hence justifying its construction and the following sections.

#### 3.1.2. Reduction to Qubits

In order to provide a clear understanding of the quantum uncertainty that arises from two measurements, it is beneficial to examine the square overlap between those measurements (Definition 4). The Bloch sphere representation, which pertains to Hilbert spaces of two dimensions, offers a lucid illustration of this concept.

Throughout this section, it is demonstrated that the devices under investigation, under specific conditions, can be expressed as a convex combination of devices, each operating on a single qubit. Working in a qubit subspace then allows one to make definitive statements regarding the entropy of the measurement outcomes produced by the device. We remark that this is in complete analogy with the proof techniques used when studying DI protocols in the non-local setting, in which one reduces the analysis to that of two single qubit devices [33] (Lemma 1).

**Lemma** **11.***Let $\tilde{D}=\left(\phi ,\tilde{\Pi },\tilde{M}\right)$ be a simplified device (Definition 14), acting within a Hilbert space H of a countable dimension, with the additional assumption that $\tilde{\Pi }$ consists of only two outcomes and let* Γ *be a Hermitian projection that commutes with both $\tilde{\Pi }$ and $\tilde{M}$. Given an operator $F=f(\tilde{M},\Gamma )$ constructed from some non-commutative polynomial of two variables f, let $S_{\tilde{D}}=\langle {f(\tilde{M},\Gamma )}_{\phi }\rangle$ be the expectation value of F.*
*Then, there exists a set of Hermitian projections ${\left\{B_{j}\right\}}_{j}$, acting within the same Hilbert space H, satisfying the following conditions*

(16)
$$\begin{array}{c} \left(1\right)\;\forall j,\;\mathrm{Rank}\left(B_{j}\right)\leq 2\\ \left(2\right)\;\sum_{j}B_{j}=1\\ \left(3\right)\;\forall j,\;\left[\tilde{\Pi },B_{j}\right]=\left[\tilde{M},B_{j}\right]=\left[\Gamma ,B_{j}\right]=0 \end{array}$$

*such that*

$$S_{\tilde{D}}=\sum_{j}\mathrm{Pr}\left(j\right)S_{{\tilde{D}}_{j}}\;,$$

*where $\mathrm{Pr}\left(j\right)=\mathrm{Tr}(B_{j}\phi )$, and $S_{{\tilde{D}}_{j}}$ is the expectation value of F given the state $\frac{B_{j}\phi B_{j}}{\mathrm{Tr}[B_{j}\phi B_{j}]}$, corresponding to the simplified device ${\tilde{D}}_{j}=\left(\frac{B_{j}\phi B_{j}}{\mathrm{Tr}[B_{j}\phi B_{j}]},\tilde{\Pi },\tilde{M}\right)$. That is, $S_{{\tilde{D}}_{j}}={\langle f\left(\tilde{M},\Gamma \right)\rangle }_{B_{j}\phi B_{j}/\mathrm{Tr}[B_{j}\phi B_{j}]}$.*


**Proof.** As an immediate result of Lemma 3, there exists a basis in which $\tilde{\Pi },\tilde{M}$ and Γ are a 2×2 block diagonal. In this basis, we take the projection on every block *j* as $B_{j}$. This satisfies the conditions in Conditions (Equation 16). Furthermore,$$S_{\tilde{D}}=\mathrm{Tr}(\sum_{j}FB_{j}\phi B_{j})=\sum_{j}\mathrm{Pr}\left(j\right)\mathrm{Tr}\left(F\frac{B_{j}\phi B_{j}}{\mathrm{Tr}(B_{j}\phi B_{j})}\right)=\sum_{j}\mathrm{Pr}\left(j\right)S_{{\tilde{D}}_{j}}\;.$$□

Note that the simplified device ${\tilde{D}}_{j}=\left(\frac{B_{j}\phi B_{j}}{\mathrm{Tr}[B_{j}\phi B_{j}]},\tilde{\Pi },\tilde{M}\right)$ yields the same expectation values to those of the simplified device ${\tilde{D}}_{j}=\left(\frac{B_{j}\phi B_{j}}{\mathrm{Tr}[B_{j}\phi B_{j}]},B_{j}\tilde{\Pi }B_{j},B_{j}\tilde{M}B_{j}\right)$. The resulting operation is therefore, effectively, performed in a space of a single qubit. In addition, due to the symmetry of $\tilde{\Pi }$ and $\tilde{M}$ in this proof, the lemma also holds for an observable constructed from $\tilde{\Pi }$ and Γ instead of $\tilde{M}$ and Γ, i.e., $S_{\tilde{D}}={f(\tilde{\Pi },\Gamma )}_{\phi }$.

The simplified protocol permits the use of the uncertainty principle, appearing in Lemma 5, in a vivid way since it has a geometrical interpretation on the Bloch sphere; recall Figure 3. Under the assumption that $\tilde{\Pi }$ has two outcomes (this has yet to be justified) we can represent $\tilde{\Pi }$ and $\tilde{M}$ as two Bloch vectors with some angle between them that corresponds to their square overlap—the smaller the square overlap, the closer the angle is to π/2. In the ideal case, the square overlap is 1/2, which means that in some basis, the two measurements are the standard and the Hadamard measurements. If one is able to confirm that $\mathrm{Pr}(\tilde{M}=0)=1$, the only possible distribution on the outcomes of $\tilde{\Pi }$ is a uniform one, which has the maximal entropy.

$\tilde{\Pi }$, however, has three outcomes and not two, rendering the reduction to qubits unjustified. We can nonetheless argue that the state being used in the protocol is very close to some other state which produces only the first two outcomes. The entropy of both states can then be related using Equation (Equation 6). The ideas described here, are combined and explained thoroughly in the main proof shown in the following subsection.

#### 3.1.3. Conditional Entropy Bound

The main proof of this subsection is performed with respect to a simplified device constructed from a standard one using Definition 15.

Before proceeding with the proof, we define the winning probability in both challenges.

**Definition** **16.**
*Given a simplified device (with implicit key k) $\tilde{D}=\left(\phi ,\tilde{\Pi },\tilde{M}\right)$, for a given y∈Y, we define the ${\tilde{\Pi }}_{y}$ and ${\tilde{M}}_{y}$ winning probabilities, respectively, as*

$$\omega_{p}^{y}:=\frac{\mathrm{Tr}(\left(1-{\tilde{\Pi }}_{y}^{2}\right)\phi_{y})}{\mathrm{Tr}(\phi_{y})}\;\;;\;\omega_{m}^{y}:=\frac{\mathrm{Tr}({\tilde{M}}_{y}^{0}\phi_{y})}{\mathrm{Tr}(\phi_{y})}\;.$$

*Likewise, the winning probabilities of* Π *and M as*
$$\omega_{p}:=\sum_{y}\mathrm{Tr}\left(\left(1-{\tilde{\Pi }}_{y}^{2}\right)\phi_{y}\right)\;\;;\;\omega_{m}:=\sum_{y}\mathrm{Tr}({\tilde{M}}_{y}^{0}\phi_{y})\;.$$
*Recall that $\phi_{y}$ are sub-normalized.*


We are now ready to prove our main technical lemma.

**Lemma** **12.**
*Let $\tilde{D}=\left(\phi ,\tilde{\Pi },\tilde{M}\right)$ be a simplified device as in Definition 14, constructed from an efficient general device $D=\left(\phi ,\Pi ,M\right)$ in the manner depicted in Definition 15. Let $\Phi_{y}\in H_{D}⊗H_{E}$ be a purification of $\phi_{y}$ (respecting the sub-normalization of $\phi_{y}$) and $\Phi ={\left\{\Phi_{y}\right\}}_{y\in Y}$. For all c∈(1/2,1] and some negligible function ξ(λ), the following inequality holds:*

(17)
$$\begin{array}{cc} H{\left(\tilde{\Pi }|E,Y\right)}_{\Phi }\geq \; & \max\{0,1-\sqrt{2}A\left(c\right)\sqrt[{4}]{1-\omega_{p}}-A\left(c\right)\sqrt{1-\omega_{m}}\}\\ \; & \;\times (\log_{2}\left(1/c\right)-h\left(\omega_{m}-2\sqrt{1-\omega_{p}}-\sqrt{2}A\left(c\right)\sqrt[{4}]{1-\omega_{p}}-A\left(c\right)\sqrt{1-\omega_{m}}\right))\\ \; & \;-\sqrt{1-\omega_{p}}\log_{2}\left(3\right)-\left(1+\sqrt{1-\omega_{p}}\right)h\left(\frac{\sqrt{1-\omega_{p}}}{1+\sqrt{1-\omega_{p}}}\right)-\xi \left(\lambda \right)\;, \end{array}$$

*where h(·) is the binary entropy function and*

(18)
$$A\left(c\right):=10/{\left(2c-1\right)}^{2}\;.$$



**Proof.** For every y∈Y, we introduce the state$$\Psi_{y}:=\frac{\left({\tilde{\Pi }}_{y}^{0}+{\tilde{\Pi }}_{y}^{1}\right)\Phi_{y}\left({\tilde{\Pi }}_{y}^{0}+{\tilde{\Pi }}_{y}^{1}\right)}{\mathrm{Tr}(\left({\tilde{\Pi }}_{y}^{0}+{\tilde{\Pi }}_{y}^{1}\right)\Phi_{y})}$$
in order to reduce the problem to a convex combination of two-dimensional ones as described in Section 3.1.2. We provide a lower bound for the entropy $H{\left(\tilde{\Pi }|E,Y\right)}_{\Psi_{y}}$ and by the continuity of entropies, a lower bound for $H{\left(\tilde{\Pi }|E,Y\right)}_{\Phi_{y}}$ is then derived. The proof proceeds in steps.
Let$$U_{y}^{0}={\tilde{\Pi }}_{y}^{0}-\left({\tilde{\Pi }}_{y}^{1}+{\tilde{\Pi }}_{y}^{2}\right)\;;\;U_{y}^{1}={\tilde{M}}_{y}^{0}-{\tilde{M}}_{y}^{1}\;.$$Using Jordan’s lemma, Lemma 2, there exists an orthonormal basis where the operators are 2×2-block diagonal. Let $\Gamma_{y}$ be the Hermitian projection on blocks where the square overlap of $U_{y}^{0}$ and $U_{y}^{1}$ is bound by *c* (good blocks). Note that $\Gamma_{y}$ also commutes with both unitaries.Using Lemma 8 we bound the probability to be in subspace where the square overlap of Π and *M* is larger than *c* (bad blocks).(19)$$\mathrm{Tr}\left(\left(1-\Gamma_{y}\right)\Psi_{y}\right)\leq \frac{2\mu +10\sqrt{\mathrm{Tr}({\tilde{M}}_{y}^{1}\Psi_{y})}}{{\left(2c-1\right)}^{2}}=A\left(c\right)\left(\frac{1}{5}\mu +\sqrt{\mathrm{Tr}({\tilde{M}}_{y}^{1}\Psi_{y})}\right)\;,$$
where A(c) is given by Equation (Equation 18) and$$\begin{array}{cc} \mu & :=|\frac{1}{2}-\mathrm{Tr}\left({\tilde{M}}_{y}^{0}{\tilde{\Pi }}_{y}^{0}\Psi_{y}{\tilde{\Pi }}_{y}^{0}\right)-\mathrm{Tr}\left({\tilde{M}}_{y}^{0}{\tilde{\Pi }}_{y}^{1}\Psi_{y}{\tilde{\Pi }}_{y}^{1}\right)|\\ \; & =\frac{1}{2}\left|\sum_{b\in \left\{0,1\right\}}\mathrm{Tr}\left({\tilde{\Pi }}_{y}^{b}\Psi_{y}{\tilde{\Pi }}_{y}^{b}\right)-2\sum_{b\in \left\{0,1\right\}}\mathrm{Tr}\left({\tilde{M}}_{y}^{0}{\tilde{\Pi }}_{y}^{b}\Psi_{y}{\tilde{\Pi }}_{y}^{b}\right)\right|\\ \; & =\frac{1}{2}\left|\sum_{b\in \left\{0,1\right\}}\mathrm{Tr}\left({\tilde{M}}_{y}^{0}{\tilde{\Pi }}_{y}^{b}\Psi_{y}{\tilde{\Pi }}_{y}^{b}\right)-\sum_{b\in \left\{0,1\right\}}\mathrm{Tr}\left(\left(1-{\tilde{M}}_{y}^{0}\right){\tilde{\Pi }}_{y}^{b}\Psi_{y}{\tilde{\Pi }}_{y}^{b}\right)\right|\;. \end{array}$$Due to Lemma 10 and Corollary 2, μ is negligible in the security parameter λ. Thus, for some negligible function η,(20)$$\mathrm{Tr}\left(\left(1-\Gamma_{y}\right)\Psi_{y}\right)\leq A\left(c\right)\sqrt{\mathrm{Tr}({\tilde{M}}_{y}^{1}\Psi_{y})}+\eta \left(\lambda \right)\;.$$Note that Φ is the state of the device (and the adversary), not Ψ. Hence, we cannot, a priori, relate $\mathrm{Tr}({\tilde{M}}_{y}^{1}\Psi_{y})$ to the winning probabilities of the device. We therefore want to translate Equation (Equation 20) to the quantities observed in the application of Protocol 3 when using the simplified device $\tilde{D}$. That is, we would like to use the values $\omega_{p},\omega_{m}$ given in Definition 16 in our equations:$$\begin{array}{cc} \sqrt{\mathrm{Tr}({\tilde{M}}_{y}^{1}\Psi_{y})} & =\sqrt{\mathrm{Tr}\left({\tilde{M}}_{y}^{1}\left(\Psi_{y}-\Phi_{y}\right)\right)+\mathrm{Tr}\left({\tilde{M}}_{y}^{1}\Phi_{y}\right)}\\ \; & \leq \sqrt{\mathrm{Tr}(|\Psi_{y}-\Phi_{y}|)}+\sqrt{\mathrm{Tr}({\tilde{M}}_{y}^{1}\Phi_{y})}\\ \; & \leq \sqrt{2\sqrt{\mathrm{Tr}({\tilde{\Pi }}_{y}^{2}\Phi_{y})}}+\sqrt{\mathrm{Tr}({\tilde{M}}_{y}^{1}\Phi_{y})}\\ \; & =\sqrt{2\sqrt{1-\omega_{p}^{y}}}+\sqrt{1-\omega_{m}^{y}}\;, \end{array}$$
where in the last inequality, we use the gentle measurement lemma [31] (Lemma 9.4.1). The last equation, combined with Equation (Equation 20), immediately yields(21)$$\mathrm{Tr}\left(\Gamma_{y}\Psi_{y}\right)\geq 1-A\left(c\right)\left(\sqrt{2}\sqrt[{4}]{1-\omega_{p}^{y}}+\sqrt{1-\omega_{m}^{y}}\right)-\eta \left(\lambda \right)\;.$$In a similar manner, for later use, we must find a lower bound on $\mathrm{Tr}({\tilde{M}}_{y}^{0}\Psi_{y})$ using quantities that can be observed from the simplified protocol. To that end, we again use the gentle measurement lemma:$$\begin{array}{cc} \mathrm{Tr}({\tilde{M}}_{y}^{0}\Phi_{y}) & =\mathrm{Tr}\left({\tilde{M}}_{y}^{0}\left(\Phi_{y}-\Psi_{y}\right)\right)+\mathrm{Tr}\left({\tilde{M}}_{y}^{0}\Psi_{y}\right)\\ \; & \leq 2\sqrt{\left(1-{\tilde{\Pi }}_{y}^{2}\right)\Phi_{y}}+\mathrm{Tr}\left({\tilde{M}}_{y}^{0}\Psi_{y}\right)\;. \end{array}$$Using the definitions of $\omega_{m}^{y},\omega_{p}^{y}$,(22)⇒ωmy≤21−ωpy+Tr(M˜y0Ψy)⇒Tr(M˜y0Ψy)≥ωmy−21−ωpy.We proceed by providing a bound on the conditional entropy, given that $\Gamma_{y}=0$, using the uncertainty principle in Lemma 5. Conditioned on being in a good subspace (which happens with probability $\mathrm{Pr}(\Gamma_{y}=0)$), the square overlap of $\tilde{\Pi }$ and $\tilde{M}$ is upper bounded by *c*. We can then bound the entropy of $\tilde{\Pi }$ using the entropy of $\tilde{M}$:$$\begin{array}{cc} H{\left(\tilde{\Pi }|E,Y=y,\Gamma =0\right)}_{\Psi_{y}} & \geq \log_{2}\left(1/c\right)-H{\left(\tilde{M}|Y=y,\Gamma =0\right)}_{\Psi_{y}}\\ \; & =\log_{2}\left(1/c\right)-h\left(\mathrm{Pr}{\left({\tilde{M}}_{y}=0|\Gamma_{y}=0\right)}_{\Psi_{y}}\right)\;. \end{array}$$Since we have a lower bound on $\mathrm{Tr}({\tilde{M}}_{y}^{0}\Psi_{y})$, we proceed by working in the regime where the binary entropy function is strictly decreasing. To that end, it is henceforth assumed that the argument of the binary entropy function, and all of its subsequent lower bounds, are larger than 1/2. By using the inequality(23)$$\mathrm{Pr}\left({\tilde{M}}_{y}=0|\Gamma_{y}=0\right)\geq \mathrm{Pr}\left({\tilde{M}}_{y}=0\right)+\mathrm{Pr}\left(\Gamma_{y}=0\right)-1\;,$$
we obtain$$-h\left(\mathrm{Pr}\left({\tilde{M}}_{y}=0|\Gamma_{y}=0\right)\right)\geq -h\left(\mathrm{Pr}\left({\tilde{M}}_{y}=0\right)+\mathrm{Pr}\left(\Gamma_{y}=0\right)-1\right)\;.$$Therefore,(24)$$H{\left(\tilde{\Pi }|E,Y=y,\Gamma =0\right)}_{\Psi_{y}}\geq \log_{2}\left(1/c\right)-h\left(\mathrm{Pr}\left({\tilde{M}}_{y}=0\right)+\mathrm{Pr}\left(\Gamma_{y}=0\right)-1\right)\;.$$We now want to bound the value $H{\left(\tilde{\Pi }|E,Y\right)}_{\Psi_{y}}$, i.e., without conditioning on the event $\Gamma_{y}=0$. We write,$$\begin{array}{cc} H{\left(\tilde{\Pi }|E,Y=y\right)}_{\Psi_{y}} & \geq H{\left(\tilde{\Pi }|E,Y=y,\Gamma_{y}\right)}_{\Psi_{y}}\\ \; & =\mathrm{Pr}\left(\Gamma_{y}=0\right)H{\left(\tilde{\Pi }|E,Y=y,\Gamma_{y}=0\right)}_{\Psi_{y}}+\\ \; & \;\;\;\mathrm{Pr}\left(\Gamma_{y}=1\right)H{\left(\tilde{\Pi }|E,Y=y,\Gamma_{y}=1\right)}_{\Psi_{y}}\\ \; & \geq \mathrm{Pr}\left(\Gamma_{y}=0\right)H{\left(\tilde{\Pi }|E,Y=y,\Gamma_{y}=0\right)}_{\Psi_{y}}\;. \end{array}$$This allows us to use Equations (Equation 21) and (Equation 24) to bound each term, respectively, on the right-hand side of the last inequality withH(Π˜|E,Y=y)Ψy≥(1−A(c)(21−ωpy4+1−ωmy)−η(λ))≥×(log2(1/c)−h(Pr(M˜y=0)+Pr(Γy=0)−1)).We use Equation (Equation 21) a second time together with Equation (Equation 22) to lower bound both probability terms in the argument of the binary entropy function and get$$\begin{array}{c} \begin{array}{cc} H{\left(\tilde{\Pi }|E,Y=y\right)}_{\Psi_{y}}\; & \geq (1-A\left(c\right)\left(\sqrt{2}\sqrt[{4}]{1-\omega_{p}^{y}}+\sqrt{1-\omega_{m}^{y}}\right)-\eta \left(\lambda \right))\times \\ \; & \;(\log_{2}\left(1/c\right)-h(\omega_{m}^{y}-2\sqrt{1-\omega_{p}^{y}}-\sqrt{2}A\left(c\right)\sqrt[{4}]{1-\omega_{p}^{y}}\\ \; & \;\;-A\left(c\right)\sqrt{1-\omega_{m}^{y}}-\eta \left(\lambda \right)))\;. \end{array} \end{array}$$Now, taking the expectation over *y* on both sides of the inequality and using Lemma 4:(25)$$\begin{array}{c} \begin{array}{cc} H{\left(\tilde{\Pi }|E,Y\right)}_{\Psi }\; & \geq (1-\sqrt{2}A\left(c\right)\sqrt[{4}]{1-\omega_{p}}-A\left(c\right)\sqrt{1-\omega_{m}}-\eta \left(\lambda \right))\times \\ \; & \;(\log_{2}\left(1/c\right)-h(\omega_{m}-2\sqrt{1-\omega_{p}}-\sqrt{2}A\left(c\right)\sqrt[{4}]{1-\omega_{p}}\\ \; & \;\;-A\left(c\right)\sqrt{1-\omega_{m}}-\eta \left(\lambda \right)))\;. \end{array} \end{array}$$Using Equation (Equation 2), we can extract η(λ) from the argument of the binary entropy function in Equation (Equation 25) such that for some negligible function ξ(λ),(26)$$\begin{array}{c} \begin{array}{cc} H{\left(\tilde{\Pi }|E,Y\right)}_{\Psi }\; & \geq (1-\sqrt{2}A\left(c\right)\sqrt[{4}]{1-\omega_{p}}-A\left(c\right)\sqrt{1-\omega_{m}})\times \\ \; & \;(\log_{2}\left(1/c\right)-h(\omega_{m}-2\sqrt{1-\omega_{p}}-\sqrt{2}A\left(c\right)\sqrt[{4}]{1-\omega_{p}}\\ \; & \;\;-A\left(c\right)\sqrt{1-\omega_{m}}))-\xi \left(\lambda \right)\;. \end{array} \end{array}$$Using the continuity bound in Equation (Equation 7) with ${∥\Psi -\Phi ∥}_{1}/2\leq \sqrt{1-\omega_{p}}$ yields(27)$$\begin{array}{cc} \left|H{\left(\tilde{\Pi }|E,Y\right)}_{\Psi }-H{\left(\tilde{\Pi }|E,Y\right)}_{\Phi }\right|\; & \leq \sqrt{1-\omega_{p}}\log_{2}\left(3\right)\\ \; & \;+\left(1+\sqrt{1-\omega_{p}}\right)\;h\left(\frac{\sqrt{1-\omega_{p}}}{1+\sqrt{1-\omega_{p}}}\right)\;. \end{array}$$Combining Equation (Equation 26) with Equation (Equation 27), we conclude that the lemma holds.
□

For brevity, denote the bound in Equation (Equation 17) as$$H\left(\tilde{\Pi }|Y,E\right)\geq g\left(\omega_{p},\omega_{m},c\right)-\xi \left(\lambda \right)\;.$$

Seeing that this inequality holds for all values c∈(1/2,1], for each winning probability pair $\left(\omega_{p},\omega_{m}\right)$ we can pick an optimal value of *c* to maximize the inequality. We do this implicitly and rewrite the bound as(28)$$g\left(\omega_{p},\omega_{m}\right):=\max_{c\in (1/2,1]}g\left(\omega_{p},\omega_{m},c\right)\;.$$

Doing so yields the graph in Figure 4a.

In the protocol, later on, the verifier chooses whether to abort or not, depending on the overall winning probability ω: $$\begin{array}{c} \begin{array}{cc} \omega & :=\mathrm{Pr}(W=1)\\ & =\mathrm{Pr}(T=0)\mathrm{Pr}(W=1|T=0)+\mathrm{Pr}(T=1)\mathrm{Pr}(W=1|T=1)\\ & =\underset{1-\beta }{\underset{︸}{\mathrm{Pr}(T=0)}}\cdot \omega_{p}+\underset{\beta }{\underset{︸}{\mathrm{Pr}(T=1)}}\cdot \omega_{m}\;. \end{array} \end{array}$$

We therefore define for every β∈(0,1), another bound on the entropy, that depends only on ω (assuming ω≥1/2) as(29)$$g\left(\omega ;\beta \right):=\min_{\omega_{p}}g\left(\omega_{p},\omega_{m}=\frac{\omega -\left(1-\beta \right)\omega_{p}}{\beta }\right)\;.$$

The optimal β can be found numerically. We plot the bound in Equation (Equation 29) in Figure 4b (neglecting the negligible element ξ(λ)).

### 3.2. Entropy Accumulation

Combining Lemma 12 with Corollary 3, we now hold a lower bound for the von Neumann entropy of Protocol 3—we can connect any winning probability ω to a lower bound on the entropy of the pre-image test. This allows us to proceed with the task of entropy accumulation. To lower bound the total amount of smooth min-entropy accumulated throughout the entire execution of Protocol 1, we use the entropy accumulation theorem (EAT), stated as Theorem 1.

To use the EAT, we need to first define the channels corresponding to Protocol 1 followed by a proof that they are in fact EAT channels. In the notation of Definition 6, we make the following choice of channels:$$M_{i}:R_{i-1}\to R_{i}\;\underset{O_{i}}{\underset{︸}{{\hat{\Pi }}_{i}{\hat{M}}_{i}}}\;\underset{S_{i}}{\underset{︸}{K_{i}T_{i}G_{i}}}\;,$$
and set $Q_{i}=W_{i}$.

**Lemma** **13.**
*The channels ${\left\{M_{i}:R_{i-1}\to R_{i}{\hat{\Pi }}_{i}{\hat{M}}_{i}K_{i}T_{i}G_{i}\right\}}_{i\in [n]}$ defined by the CPTP map describing the i-th round of Protocol 1 as implemented by the computationally bounded untrusted device $\tilde{D}$ and the verifier are EAT channels according to Definition 6.*


**Proof.** To prove that the constructed channels ${\left\{M_{i}\right\}}_{i\in [n]}$ are EAT channels, we need to show that the conditions are fulfilled:
${\left\{O_{i}\right\}}_{i\in [n]}={\{{\hat{\Pi }}_{i}\}{\hat{M}}_{i}}_{i\in [n]}$, ${\left\{S_{i}\right\}}_{i\in [n]}={\left\{K_{i}T_{i}G_{i}\right\}}_{i\in [n]}$ and ${\left\{Q_{i}\right\}}_{i\in [n]}={\left\{W_{i}\right\}}_{i\in [n]}$ are all finite-dimensional classical systems. ${\left\{R_{i}\right\}}_{i\in [n]}$ are arbitrary quantum systems. Finally, we have$$d_{O}=d_{{\hat{\Pi }}_{i}}\cdot d_{{\hat{M}}_{i}}=\left|\left\{0,1,2\right\}\right|\cdot \left|\left\{0,1\right\}\right|=6<\infty \;.$$For any i∈[n] and any input state $\sigma_{R_{i-1}}$, $W_{i}$ is a function of the classical values ${\hat{\Pi }}_{i}{\hat{M}}_{i}K_{i}T_{i}G_{i}$. Hence, the marginal $\sigma_{O_{i}S_{i}}$ is unchanged when deriving $W_{i}$ from it.For any initial state $\rho_{R_{0}E}^{in}$ and the resulting final state $\rho_{\mathrm{OSQ}E}=\rho_{\Pi MKTGWE}$, the Markov chain conditions$${\left(\hat{\Pi }\hat{M}\right)}_{1},\cdots ,{\left(\hat{\Pi }\hat{M}\right)}_{i-1}↔{\left(KTG\right)}_{1},\cdots ,{\left(KTG\right)}_{i-1},E↔{\left(KTG\right)}_{i}$$
trivially hold for all i∈[n], as $K_{i},T_{i}$ and $G_{i}$ are chosen independently from everything else.
□

**Lemma** **14.***Let β,ω,γ∈(0,1) and g(ω;β) be the function in Equation* (Equation 29). *Let p be a probability distribution over W={⊥,0,1} such that γ=1−p(⊥) and ω=p(1)/γ. Define $\Sigma \left(p\right)=\{\sigma_{R_{i-1}R^{\prime }}:M_{i}{\left(\sigma \right)}_{W_{i}}=p\}$. Then, there exists a negligible function ξ(λ) such that*
(30)$$\left(g\left(\omega ;\beta \right)-\xi \left(\lambda \right)\right)\left(1-\beta \gamma \right)\leq \underset{\sigma_{R_{i-1}R^{\prime }}\in \Sigma \left(p\right)}{\inf}H{\left({\hat{\Pi }}_{i}{\hat{M}}_{i}|K_{i}T_{i}G_{i}R^{\prime }\right)}_{M_{i}\left(\sigma \right)}\;.$$
*In particular, this implies that for every β∈(0,1), the function*

(31)
$$f_{\min}:=\left(p\right)\left(g\left(\omega ;\beta \right)-\xi \left(\lambda \right)\right)\left(1-\beta \gamma \right)$$

*satisfies Definition 7 and is therefore a min-trade-off function.*


**Proof.** Due to Lemma 12 and the consequent Equation (Equation 29), the following holds for any polynomial-sized state σ (not necessarily efficient):g(ω;β)−ξ(λ)≤H(Π^i|YiKi,Ti=0,R′)Mi(σ)≤1Pr(Ti=0)Pr(Ti=0)H(Π^i|YiKi,Ti=0,R′)Mi(σ)=1Pr(Ti=0)+Pr(Ti=1)H(Π^i|YiKi,Ti=1,R′)Mi(σ)=1Pr(Ti=0)H(Π^i|YiKiTiR′)Mi(σ)=11−βγH(Π^i|YiKiTiR′)Mi(σ)≤11−βγH(Π^iM^i|KiTiR′)Mi(σ)=11−βγH(Π^iM^i|KiTiGiR′)Mi(σ).For the last equality, note that the device only knows which test to perform while being unaware if it is for a generation round or not. Therefore, once *T* is given, *G* does not provide any additional information. □

Using Theorem 1, we can bound the smooth min-entropy resulting in the application of Protocol 1: (32)$$H_{\min}^{\varepsilon_{s}}{\left(\hat{\Pi }\hat{M}|\mathrm{KTG}\right)}_{\rho_{|\Omega }}\geq nf_{\min}-\mu \sqrt{n}\;,$$
where $f_{\min}$ is given by Equation (Equation 31). We simplify the right-hand side in Equation (Equation 32) to a single entropy accumulation rate $\mu_{\mathrm{opt}}$, and a negligible reduction ξ(λ): (33)$$H_{\min}^{\varepsilon }{\left(\hat{\Pi }\hat{M}|KTG\right)}_{\rho_{|\Omega }}\geq n\left(\mu_{\mathrm{opt}}\left(n,\omega ,\gamma ,\varepsilon_{s},p_{\Omega };\beta \right)-\xi \left(\lambda \right)\right)\;.$$

Note that the differential of $f_{\min}$ is unbounded. This prevents us from using the EAT due to Equation (Equation 14). This issue is addressed by defining a new min-trade-off function ${\tilde{f}}_{\min}$ such that(34)$${\tilde{f}}_{\min}\left(\omega ;\omega_{0}\right)=\left\{\begin{array}{cc} f_{\min}\left(\omega \right) & \omega \leq \omega_{0}\\ \frac{d}{d\omega }{\left.f_{\min}\left(\omega \right)\right|}_{\omega =\omega_{0}}\left(\omega -\omega_{0}\right)+f_{\min}\left(\omega_{0}\right) & \mathrm{otherwise} \end{array}\right.\;.$$

We provide a number of plots for $\mu_{\mathrm{opt}}$ as a function of ω for various values of *n* in Figure 5. We remark that we did not fully optimize the code to derive the plots, and one can probably derive tighter plots.

### 3.3. Randomness Rates

In the previous subsection, we derive a lower bound on $H_{\min}^{\varepsilon }{\left(\hat{\Pi }\hat{M}|KTG\right)}_{\rho_{|\Omega }}$. However, our ultimate goal is to provide a bound on the designated source of randomness, namely $H_{\min}^{\varepsilon_{s}}{\left(\hat{\Pi }|\mathrm{KTG}E\right)}_{\rho_{|\Omega }}$. The following result, Theorem 2, formalizes this guarantee and constitutes the main theorem of this work. It gives a concrete lower bound on the min-entropy of the protocol’s output, conditioned on the adversary’s information, under a computational assumption on the device. While Theorem 1 provides the general entropy accumulation framework applicable to our setting, the core technical difficulty lies in establishing the single-round von Neumann entropy bound proved in Lemma 12. This bound captures the contribution of the cryptographic structure and enables the application of Theorem 1, ultimately leading to the main result in Theorem 2.

**Theorem** **2.***Let D be a device executing Protocol 1, and let ρ denote the joint quantum state generated at the end of the protocol. Let ω∈[0,1] be a fixed threshold specifying the minimum fraction of successfully passed equation test rounds required for the verifier not to abort. Let* Ω *denote the event that the device achieves a test score of at least ω on the test rounds, and define $p_{\Omega }:=\mathrm{Pr}\left[\Omega \right]$. (This dependence on $p_{\Omega }$ reflects a known limitation of the entropy accumulation theorem: the bound applies only conditionally, and the probability of passing the test (i.e., $p_{\Omega }$) is not provided by the theorem itself. However, in practice, $p_{\Omega }$ can be estimated or lower bounded using concentration inequalities.) Let $\rho_{|\Omega }$ denote the state ρ conditioned on the event* Ω, *and let $\varepsilon_{s}>0$ be a smoothness parameter. Then, the following bound holds:*
(35)Hminεs(Π^∣KTGE)ρ|Ω≥nμopt(n,ω,γ,εs/4,pΩ;β)−ξ(λ)−γ≥−2log(7)1−2logεs4·pΩ−3log1−1−(εs/4)2.

**Proof.** We begin with entropy chain rule [34] (Theorem 6.1)$$\begin{array}{c} \begin{array}{cc} H_{\min}^{\varepsilon_{s}}{\left(\hat{\Pi }|\mathrm{KTG}E\right)}_{\rho_{|\Omega }} & \geq H_{\min}^{\frac{\varepsilon_{s}}{4}}{\left(\hat{\Pi }\hat{M}|\mathrm{KTG}E\right)}_{\rho_{|\Omega }}-H_{\max}^{\frac{\varepsilon_{s}}{4}}{\left(\hat{M}|\mathrm{KTG}E\right)}_{\rho_{|\Omega }}\\ \; & \;\;\;-3\log(1-\sqrt{1-{\left(\varepsilon_{s}/4\right)}^{2}})\;. \end{array} \end{array}$$The first term on the right-hand side is given in Equation (Equation 33); it remains to find an upper bound for the second term. Let us start from$$H_{\max}^{\frac{\varepsilon_{s}}{4}}{\left(\hat{M}|\mathrm{KGT}E\right)}_{\rho_{|\Omega }}\leq H_{\max}^{\frac{\varepsilon_{s}}{4}}{\left(\hat{M}|TE\right)}_{\rho_{|\Omega }}\;.$$We then use the EAT again in order to bound $H_{\max}^{\frac{\varepsilon_{s}}{4}}{\left(\hat{M}|TE\right)}_{\rho_{|\Omega }}$. We identify the EAT channels with $O\to \hat{M},S\to T$ and E→E. The Markov conditions then trivially hold, and the max-trade-off function reads$$f_{\max}\left(p\right)\geq \underset{\sigma_{R_{i-1}R^{\prime }}:M_{i}{\left(\sigma \right)}_{W_{i}}=\omega }{\sup}H{\left({\hat{M}}_{i}|T_{i}R^{\prime }\right)}_{M_{i}\left(\sigma \right)}\;.$$Since the following distributions are satisfied for all i∈[n]
$$\begin{array}{cc} \mathrm{Pr}[{\hat{M}}_{i}=⊥|T_{i}=0] & =1,\\ \mathrm{Pr}[{\hat{M}}_{i}\in \left\{0,1\right\}|T_{i}=1] & =1,\\ \mathrm{Pr}[T_{i}=1] & =\gamma , \end{array}$$
the max-trade-off function is simply $f_{\max}\left(p\right)=\gamma$ (thus ${∥\nabla f_{\max}∥}_{\infty }=0)$. We therefore get$$H_{\max}^{\frac{\varepsilon_{s}}{4}}{\left(\hat{M}|TE\right)}_{\rho_{|\Omega }}\leq \gamma n+2\log\left(7\right)\sqrt{1-2\log(\frac{\varepsilon_{s}}{4}\cdot p_{\Omega })}\;.$$□

We wish to make a final remark. One could repeat the above analysis under the assumption that *Y* and *X* are leaked, to obtain a stronger security statement (this is, however, not performed in previous works). In this case, $Y_{i}$ and $X_{i}$ should be part of the output $O_{i}$ of the channel. Then, the dimension of $O_{i}$ scales exponentially with λ, which worsens the accumulation rate $\mu_{\mathrm{opt}}$ since this results in exponential factors in Equation (Equation 13). In that case, $\mu_{\mathrm{opt}}$ becomes a monotonically decreasing function of λ and we obtain a certain trade-off between the two elements in the following expression:$$\mu_{\mathrm{opt}}\left(n,\omega ,\gamma ,\varepsilon_{s},p_{\Omega },\lambda ;\beta \right)-\xi \left(\lambda \right)\;.$$

This means that there is an optimal value of λ that maximizes the entropy and to find it, one requires an explicit bound on ξ(λ).

### 3.4. Generalization to Computational CHSH

It is worth considering a couple of recent works in which the proof techniques, introduced in the current one, may be generalized. In [14] a test of quantumness is developed for a family of protocols with a concrete example in the form of the KCVY Protocol [11]. In [15], explicit bounds are provided on compiled non-local games and more specifically, a compiled CHSH game where a classical verifier acts both as a verifier and one of the players in a standard CHSH game while the second player is a quantum device.

Both of these works use computational problems which bear a strong resemblance to the standard non-local CHSH game. Furthermore, ref [15] (Lemma 34) and ref [14] (Theorem 4.7) provide an upper bound on the anti-commutation relations of the challenges the prover receives as a function of the winning probability in the protocol—This potentially allows the use of the Entropic uncertainty principle Lemma 5. As an example, in an optimal winning probability, the anti-commutator is negligibly close to zero (it is also worth mentioning that this is a similar condition that must also be met in the standard non-local CHSH game).

Similarly to the upper bound on the projection on “bad blocks” in Equation (Equation 19), one could use the lemma to provide a similar bound using a Markov inequality. Assuming now that the subspaces we are working with have measurements whose square overlap is bound by *c*, one may use the statistics of each measurements, geometrical arguments (along with rigidity) and entropic uncertainty relations, to provide a lower bound on the entropies of each measurement. It is worth noting that even with an honest prover, the von Neumann entropy will be ≈h(0.15). This could potentially be remedied by introducing a third measurement but regardless, once a lower bound on the von Neumann entropy is found, similar steps to those of Section 3.2 can be followed.

## 4. Conclusions and Outlook

By utilizing a combination of results from quantum information theory and post-quantum cryptography, we have shown that entropy accumulation is feasible when interacting with a single device. While this was previously performed in [5] using ad hoc techniques, we provide a flexible framework that builds on well-studied tools and follows similar steps to those used in DI protocols based on the violation of Bell inequalities using two devices [27]. Prior to our work, it was believed that such an approach cannot be taken (see the discussion in [5]). We remark that while we focused on randomness *certification* in the current manuscript, one could now easily extend the analysis to randomness *expansion, amplification* and *key distribution* using the same standard techniques applied when working with two devices [19,27].

Furthermore, even though we carried out the proof here specifically for the computational challenge derived from a NTCF, the methods that we establish are modular and can be generalized to other protocols with different cryptographic assumptions. For example, in two recent works [14,15], the winning probability in various “computational games” is tied to the anti-commutator of the measurements used by the device that plays the game (see [15] (Lemma 38) and [14] (Theorem 4.7) in particular). Thus, their results can be used to derive a bound on the conditional von Neumann entropy as we do in Lemma 12. From there onward, the final bound on the accumulated smooth min-entropy is derived exactly as in our work.

Apart from the theoretical contribution, the new proof method allows us to derive explicit bounds for a finite number of rounds of the protocol, in contrast to asymptotic statements. Thus, one can use the bounds to study the practicality of DI randomness certification protocols based on computational assumptions.

For the *current protocol*, in order to obtain a positive rate, the number of repetitions *n* required, as seen in Figure 5, is too demanding for actual implementation. In addition, the necessary observed winning probability is extremely high. We pinpoint the “source of the problem” to the min-trade-off function presented in Figure 4 and provide below a number of suggestions as to how one might improve the derived bounds. In general, however, we expect that for other protocols (e.g., those suggested in [14,15]), one could derive better min-trade-off functions that will bring us closer to the regime of experimentally relevant protocols. In fact, our framework allows us to compare different protocols via their min-trade-off functions and thus can be used as a tool for benchmarking new protocols.

Beyond these directions, the suggestions raised in the review point toward further refinements that may help reduce the round complexity. In particular, one could explore variants of the entropy accumulation framework that allow for memory-aware entropy tracking or dynamically weighted contributions across rounds. Another possibility is to structure the protocol in layers, separating fast verification from more costly entropy collection. While we do not pursue these directions here, our framework is compatible with such modifications and could serve as a foundation for their analysis.

We conclude with several open questions.

In both the original analysis performed in [5] and our work, the cryptographic assumption needs to hold even when the (efficient) device has an (inefficient) “advice state”. Including the advice states is necessary when using the entropy accumulation in its current form, due to the usage of a min-trade-off function (see Equation (Equation 30)). One fundamental question is therefore whether this is *necessary* in all DI protocols based on the post-quantum computational assumptions or not.As mentioned above, what makes the protocol considered here potentially unfeasible for experimental implementations is its min-trade-off function. Ideally, one would like to both decrease the winning probability needed in order to certify entropy as well as the *derivative* of the function, which is currently too large. The derivative impacts the second-order term of the accumulated entropy, and this is why we observe the need for a large number of rounds in the protocol—many orders of magnitude more than in the DI setup with two devices. Any improvement of Lemma 12 may be useful; when looking into the details of the proof, there is indeed some room for it.Once one is interested in non-asymptotic statements and actual implementations, the unknown negligible function ξ(λ) needs to be better understood. The assumption is that ξ(λ)=0 as λ→∞, but in any given execution, one does fix a finite λ. Some more explicit statements should then be made regarding ξ(λ) and incorporated into the final bound.In the current manuscript, we worked with the EAT presented in [25]. A generalized version, that allows for potentially more complex protocols, appears in [26]. All of our lemmas and theorems can also be derived using [26] without any modifications. An interesting question is whether there are DI protocols with a single device that can exploit the more general structure of [26].

## Figures and Tables

**Figure 1 entropy-27-00772-f001:**
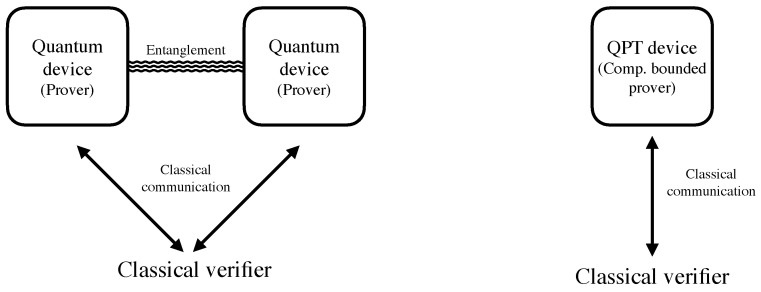
Two setups for device-independent protocols. On the left, a classical verifier is interacting classically with two non-communicating but otherwise all powerful quantum devices (also called provers) that can share entanglement. On the right, the verifier is interacting with a single polynomial-time quantum computer.

**Figure 2 entropy-27-00772-f002:**
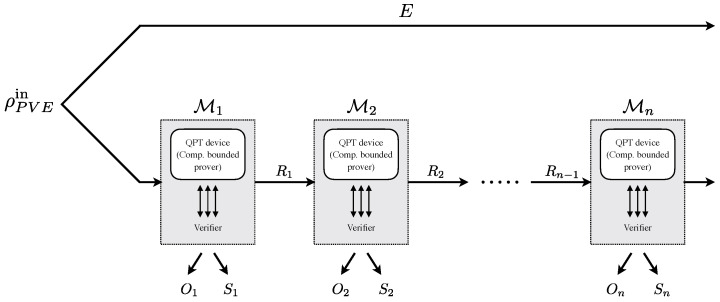
The general structure of a protocol that we consider. The initial state of the entire system is $\rho_{PVE}^{in}$. The protocol proceeds in rounds: Each round includes an interaction between the verifier and the prover, as shown by the gray boxes in the figure, and can be described by an efficient quantum channel $M_{i}$ for every round i∈[n]. The channels output outcomes $O_{i}$ and side information $S_{i}$. The device may keep the quantum memory from previous rounds using the registers $R_{i}$. The adversary’s system *E* is untouched by the protocol. This structure fits the setup of the entropy accumulation theorem [25] and can easily be extended to that of the generalized entropy accumulation [26].

**Figure 3 entropy-27-00772-f003:**
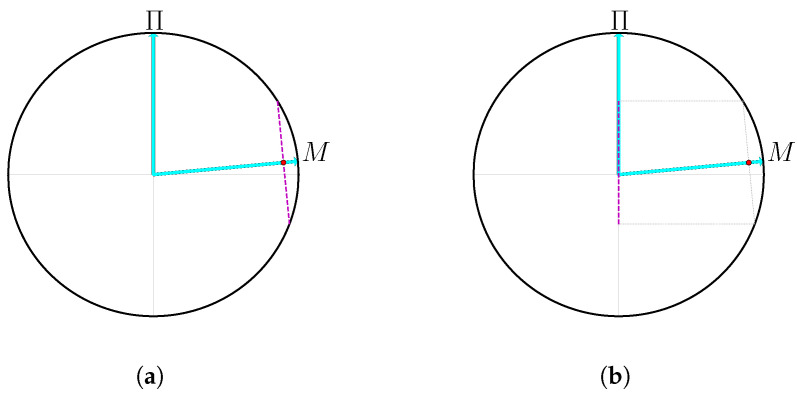
A Bloch sphere representation of the uncertainty relations. A small square overlap between Π and *M* corresponds to a larger angle between the two. Each point on the arrow *M* corresponds to a probability Pr(M=0) of some unknown state ρ that must lie on the corresponding magenta dashed line in (**a**). All the states described in (**a**) allow distributions of Pr(Π=0) that correspond to the points on the dashed magenta line described in (**b**). A small square overlap and high values of Pr(M=0) mean that the allowed distributions of Pr(Π=0) are close to uniform. (**a**) All possible states with Pr(M=0) on the magenta dashed line. (**b**) All possible distributions of Pr(Π=0) on the magenta dashed line.

**Figure 4 entropy-27-00772-f004:**
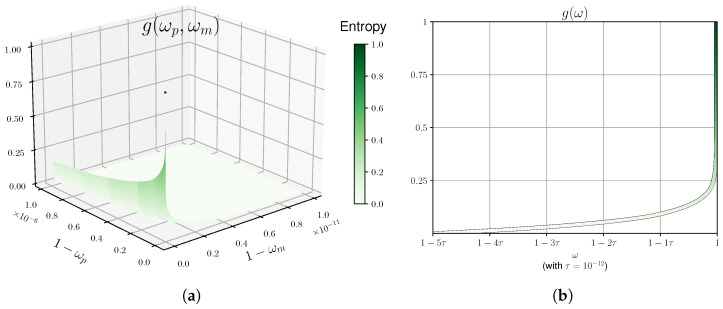
The entropy as a function of the winning probabilities as given by Equations (Equation 28) and (Equation 29). In the left plot, the differential at $\left(\omega_{p}=1,\omega_{m}=1\right)$ diverges, and therefore one does not see the entropy grow to its optimal value (the dark green point); this is only a visual effect. This entropy and the divergence are seen more clearly on a slice of this graph, appearing in the right plot. (**a**) Plot of the function given in Equation (Equation 28). (**b**) Plot of the function appearing in Equation (Equation 29) with β=0.045. The width of the curve is added for clarity.

**Figure 5 entropy-27-00772-f005:**
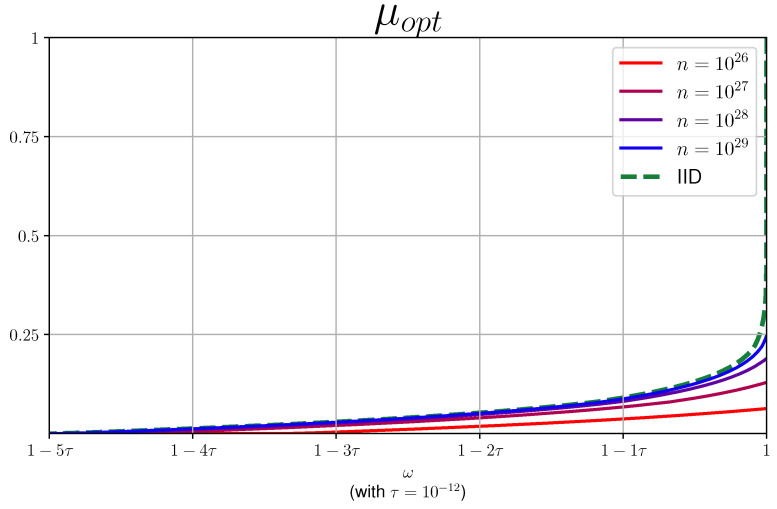
Plots of the entropy accumulation rates $\mu_{\mathrm{opt}}\left(n,\omega ,\gamma =1,\varepsilon_{s}=10^{-5},p_{\Omega }=10^{-5}\right)$ as a function of ω for various values of *n*. For reference, we include the accumulation rate of the IID case as a function of ω (Figure 4b), to which all rates converge in the limit n→∞. We neglect the negligible element in Equation (Equation 33).

## Data Availability

The original contributions presented in this study are included in the article. Further inquiries can be directed to the corresponding author.

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
