# Peer review of "Entropy Accumulation Under Post-Quantum Cryptographic Assumptions"

_entropy, 2025, doi:10.3390/e27080772_

Round 1

Reviewer 1 Report

Comments and Suggestions for Authors

This paper presented a novel framework for analyzing entropy accumulation in device-independent quantum protocols under post quantum cryptographic assumptions, this work addresses a gap by providing a modular proof technique inspired by non-local device-independent protocols. My detailed comments for are as follows:

1) The introduction is dense. It requires a more streamlined overview of the background, motivations and main contributions. For instance, a direct comparison with the limitations of previous methods would strengthen the motivation.

2) The transition between sections, particularly from the single-round analysis to the multi-round entropy accumulation, should be smoother. Some intermediate steps are assumed, which may not be immediately clear to all readers.

3) A separate related work section is required to comprehensively review state-of-the-art methods in this field and other related fields, which would facilitate the problem formulation.

4) The quantitative results indicate that the protocol requires an impractical number of rounds for meaningful entropy rates. The authors should address whether this is a fundamental limitation or an artifact of their analysis.

5) The discussion of computational CHSH is underdeveloped. A more detailed sketch of how the framework applies to these protocols would be helpful.

6) Some symbols are introduced without clear explanations. A glossary or expanded definitions would help.

7) More high-quality figures are suggested to better demonstrate the proposed method and experimental results. A good overview would help readers understand the proposed method. Please consider using more visualizations to demonstrate the effectiveness of the proposed method.

8) The experiments should include more baselines, datasets and evaluation metrics to support the proposed method.

9) A separate discussion section is needed to show the limitations of the proposed method and future directions. For instance, the authors should consider the potential combination with machine/deep learning to enhance the proposed method. Some related papers are recommended and should be included in the reference list: variance-constrained local-global modeling for device-free localization under uncertainties, IEEE TII, and leveraging online learning for domain-adaptation in wi-fi-based device-free localization, IEEE TMC.

Reviewer 2 Report

Comments and Suggestions for Authors

In the paper, the security proof of device-independent (DI) quantum randomness certification under computational post-quantum assumptions using the technique of entropy accumulation has been obtained. As the title "device-independent" suggests, this kind of cryptographic protocols and security proof impose no or only very basic assumptions on the devices and user(s) can verify that the device is 'honest' using certain tests and natural restrictions of the devices. In DI quantum key distribution (QKD), the assumption that two distant devices do not communicate (e.g., because of the impossibility of superluminal signalling) is a natural assumptions, which allow the legitimate users to check honesty of the devices.

In this paper, the problem of randomness certification, where there is only one device, is in focus. Since there is only one device, the assumption of no signalling makes no sense and we need another natural restriction on the capabilities of the device. The authors use the assumption that computational power of the device is limited by problems of (quantum) polynomial complexity. Using this, the authors adopt the known technique of entropy accumulation widely used in device-independent quantum cryptography to prove the security of quantum randomness certification. The protocol is based on the existence of noisy trapdoor claw-free functions.

The results are very interesting and definitely can be recommended for publication. The authors give detailed description of definitions, statements and proofs. However, I strongly recommend the authors to pay more attention to explanations. The paper is hard to read, especially to non-specialists in entropy accumulation technique and DI cryptography. Though the authors explain certain things, more effort in this direction is required. In order for paper to be more understandable to a broader community of quantum information, the following points can be clarified:

  1. What is the main theorem of the paper? As far as I understand, Theorem 1 is a known theorem about general technique of entropy accumulation. Then Theorem 2? But it looks too technical.
  2. The authors several times stress the development with respect to Ref. [5], where some "ad hoc techniques" are used. In order for a reader to better understand the contribution of this paper, could the authors describe it more concretely? As a reviewer, I cannot clearly understand and explain the advantage of this paper with respect to Ref. [5] based on the present manuscript. The authors understand what do they mean by "ad hoc techniques", but for a reader not very familiar with the topic this are just abstract unclear words. In line 221, the authors say that the main advancement of this work in comparison with Ref. [5] is "the ability to entirely replace [5, Section 8]". However, again, in order to understand what does it mean, a reader must address Ref. [5]. Can the authors explain more concretely what is Section [8] of Ref. [5] about and what precisely do they replace?
  3. More explanations about Protocol 1 must be given. Before a formal proof, what is the intuition why this should work? I do not understand what is the 'game' here. For example, I do not understand why the generated random bit \hat П_i has such a tricky form.
  4. Also I do not understand the difference between generation and test round when T_i=0. It seems that the verifier does the same. What precisely does allow him to test when G_i=0 and does not allow to test when G_i=0? Let me clarify what I mean. In DI QKD, in test rounds, the users choose measurements required to check the Bell inequality and, in the generation, they choose computational basis. So, some of their settings does not allow to generate the key, but allow to check the honesty of the device, while other settings allow to generate a key bit. Here I do not understand such a difference.
  5. Also I cannot understand what is the 'honest' realisation of the device D. For example, in DI QKD, an honest device distributes certain entangled states and performs specified measurements. What is the honest behaviour here, i.e., 'honest' phi, П and M in Definitions 11 and 12?
  6. What is w in Definition 11 (line 453) and where does it play a role?

General comment about all the questions: It seems that the authors, though gave some basic explanations, wrote the paper for people already familiar with the topic and especially Ref. [5]. To simplify understanding this paper and for self-consistency, more explanations must be given.

Reviewer 3 Report

Comments and Suggestions for Authors

The paper proposes a modular proof framework inspired by non-local device-independent literature techniques, combining entropy uncertainty relations and the entropy accumulation theorem in quantum information theory. Under the single-device computational scenario and based on post-quantum cryptography assumptions, it provides a conceptually clear systematic analysis and quantitative security guarantees for device-independent protocols, supporting the design and security proof of protocols such as randomness generation and expansion.

The paper currently has some formatting issues that need to be improved. Once the formatting issues are carefully revised, I believe the paper can be accepted.

  1. Key formulas lack symbolic annotations. For example, Formula (1) does not explain the physical meanings of σ and Mi. Additionally, some formulas do not have punctuation at the end, such as Formulas (5) and (9).
  2. The reference formatting is extremely non-standard, first in terms of inconsistency in style, second in significant deviation from journal requirements, and finally, some early references are still in preprint form, such as References 7 and 17.
  3. The paper's English level meets academic publication standards, with accurate language and rigorous logic. Further optimizing long sentence structures and adding popular annotations for technical terms can enhance readability for cross-disciplinary readers.

I propose some follow-up research directions for this field. Of course, the authors can also consider improving this paper, but that would require considerable effort.

  1. The second-order term loss of the Entropy Accumulation Theorem (EAT) and the unbounded derivative of the min-tradeoff function lead to the need for a large number of rounds to offset errors. To achieve a positive randomness rate, the protocol needs to perform n≈1026 rounds of interaction and requires an extremely high device test pass rate, far exceeding the capabilities of existing quantum hardware.

Suggestions: Optimize the application of EAT by adopting the *Generalized Entropy Accumulation Theorem* and using its more flexible state evolution model to reduce the number of rounds. For example, dynamically adjust the entropy accumulation weight of each round through the "quantum channel memory effect". Split single-round complex tests into a "basic verification layer" and a "randomness amplification layer", where the former quickly filters invalid devices, and the latter purposefully accumulates entropy to reduce the total number of rounds.

  1. The protocol relies on the assumption that "devices cannot solve the LWE problem with inefficient advice states", which is stronger than the standard LWE assumption (only for efficient devices), and the rationality of this assumption is not proven. If it is discovered in the future that inefficient advice states can crack LWE, the protocol's security will collapse.

Suggestions: Prove through "quantum indistinguishability" to simplify the assumption to "the hardness of the standard LWE problem for quantum polynomial-time devices", referring to the "computational indistinguishability" lemma to remove the dependency on inefficient advice states. Combine zero-knowledge proof techniques to allow devices to self-certify "not using efficient advice states" in the protocol, such as verifying the computational complexity of their internal state preparation process through interactive proofs.

  1. The key conclusions use "negligible functions", but specific expressions are not provided, making it impossible to evaluate practical security under finite security parameters. The vague mathematical treatment makes the protocol unable to pass cryptographic standard tests (such as NIST post-quantum cryptography evaluation).

Suggestions: Derive the specific form of "negligible functions" through the hardness proof of the "Shortest Vector Problem" in lattice cryptography, and clarify the security threshold in practical applications.

Round 2

Reviewer 1 Report

Comments and Suggestions for Authors

The authors did not address the issues, this paper cannot be accepted.

Author Response

We thank the reviewer for their time and for their detailed feedback on the manuscript.

We carefully reviewed each of the comments and suggestions. While we found the points to be thoughtful and constructive, we have ultimately chosen to maintain the current focus and scope of the paper as a theoretical contribution to quantum cryptography.

In particular, several of the reviewer’s suggestions, such as the inclusion of additional datasets, empirical baselines, or machine learning-based approaches, are not applicable to our work, which is entirely formal and proof-theoretic in nature.

Regarding the requests for structural changes (e.g., a separate related work section, additional figures, or a glossary), we have clarified key definitions (Such as Theorem 2 and Definiton 12) and improved transitions within the existing structure, but we believe the current format maintains coherence and readability given the nature of the content.

We remain open to further revisions if specific points require deeper clarification, and we appreciate the reviewer’s engagement with our work.

Reviewer 2 Report

Comments and Suggestions for Authors

The authors addressed my comments and improved the quality of the paper, so now I can understand it much better. However, still some things remain unclear, especially for non-specialist in this field. I recommend publication if the authors take into account the following comments (arose after better understanding after authors' clarifications):

  1. Definition 10, line 421 about the algorithm Chk_k. It is worth to write whether this algorithm requires the knowledge of the trapdoor or not (in the case of any answer). It seems that if it does not require the trapdoor, then it can be used also by device to cheat in the preimage test (not to output the not valid pairs (b_i,x_i)).
  2. The same question for a perfect matching relation in lines 415-416: Can this relation (subset) be efficiently reconstructed without the trapdoor? It is better to write in the case of any answer.
  3. Protocol 1: What is "valid pair" in line 3 (equation test)? It seems that they are related to H_k from Definition 10 between lines 432 and 433, but I cannot find where it is written explicitly.
  4. Also "valid answers" are mentioned in Lemma 10, line 516, along with a new notation V_{y,1}. Again, it is better to explain the relation of V_{y,1} to H_k or something like that.
  5. Protocol 1: I think it worth to write explicitly that, in the test round, T_i=0 and T_i=1 are the preimage test and the equation test, respectively (in the lines of the protocol or in the caption).
  6. Theorem 2: Since it is the main result, it is better to formulate it more self-consistently, in order for a reader to be able to jump to it right after studying Protocol 1: What do we study and what is the main result. The first point here is p_Omega. Then all quantities I cannot find p_Omega in the description of Protocol 1, but probably it is related to delta_est. I cannot find an exact relation as well as instances of delta_est after the description of the protocol (like it did not play any role).
  7. Also in the description of Protocol 1, we have omega_exp, which is not met after that. In Theorem 2, we have omega without a subindex instead. Is it the same?
  8. Also, for a convenience of a reader (for self-consistency of Theorem 2), I think, a reference to formula for omega between lines 675 and 676 should be given. It took some time for me to find it in order to understand the statement of Theorem 2.
